# New black indium oxide—tandem photothermal $CO_2$-$H_2$ methanol selective catalyst

Zeshu Zhang[1], Chengliang Mao [2,3✉], Débora Motta Meira [4,5], Paul N. Duchesne [6], Athanasios A. Tountas[2], Zhao Li[2], Chenyue Qiu [7], Sanli Tang[2], Rui Song[2], Xue Ding[1], Junchuan Sun[1], Jiangfan Yu [1], Jane Y. Howe[7,8], Wenguang Tu [1], Lu Wang [1✉] & Geoffrey A. Ozin [2✉]

It has long been known that the thermal catalyst $Cu/ZnO/Al_2O_3$(CZA) can enable remarkable catalytic performance towards $CO_2$ hydrogenation for the reverse water-gas shift (RWGS) and methanol synthesis reactions. However, owing to the direct competition between these reactions, high pressure and high hydrogen concentration ($\geq$75%) are required to shift the thermodynamic equilibrium towards methanol synthesis. Herein, a new black indium oxide with photothermal catalytic activity is successfully prepared, and it facilitates a tandem synthesis of methanol at a low hydrogen concentration (50%) and ambient pressure by directly using by-product CO as feedstock. The methanol selectivities achieve 33.24% and 49.23% at low and high hydrogen concentrations, respectively.

[1] School of Science and Engineering, The Chinese University of Hong Kong, Shenzhen, 518172 Shenzhen, Guangdong, People's Republic of China. [2] Solar Fuels Group, Department of Chemistry, University of Toronto, 80 St. George Street, Toronto, ON M5S 3H6, Canada. [3] Key Laboratory of Pesticide & Chemical Biology of Ministry of Education, Institute of Environmental & Applied Chemistry, College of Chemistry, Central China Normal University, Wuhan 430079, People's Republic of China. [4] CLS@APS, Advanced Photon Source, Argonne National Laboratory, Lemont, IL 60439, USA. [5] Canadian Light Source Inc., 44 Innovation Boulevard, Saskatoon, SK S7N 2V3, Canada. [6] Department of Chemistry, Queen's University, 90 Bader Lane, Kingston, ON K7L 3N6, Canada. [7] Department of Materials Science and Engineering, University of Toronto, 184 College Street, Toronto, ON M5S 3E4, Canada. [8] Department of Chemical Engineering and Applied Chemistry, University of Toronto, 200 College St, Toronto, ON M5S 3E5, Canada. ✉email: chengliang.mao@mail.utoronto.ca; lwang@cuhk.edu.cn; g.ozin@utoronto.ca

Methanol is one of the most valuable chemicals with a worldwide demand of more than 80 million metric tons in 2019[1–5]. It can be considered a clean fuel, an alternative hydrogen carrier, and an essential building block for about 30% of known chemicals[6–8]. Thus, storing solar energy in the form of renewable methanol while reducing the $CO_2$ concentration of the atmosphere could have the potential to close the carbon cycle, produce renewable fuels, and ameliorate climate change[9–16].

However, based on Le Chatelier's principle and the competitive situation of the RWGS reaction and methanol synthesis, the traditional route for methanol production usually employs a high concentration of $H_2$ and high pressure to shift the reaction equilibrium and improve methanol selectivity[17–21]. Alternatively, a tandem reaction pathway may pave the way for methanol synthesis with low pressure and low $H_2$ concentration (50%) by conducting the RWGS and CO hydrogenation reactions according to Eqs. 1 and 3[22–25]. To achieve this scenario, a catalyst with multifunctional active sites that can conduct methanol synthesis from both $CO_2$ and CO is desired, and such a material is yet to be developed.

$$CO_2 + H_2 = CO + H_2O \qquad (1)$$

$$CO_2 + 3H_2 = CH_3OH + H_2O \qquad (2)$$

$$CO + 2H_2 = CH_3OH \qquad (3)$$

Herein, a new black form of tailored nanoscale indium oxide with oxygen vacancies, hydroxyls, and hydride sites $H_zIn_2O_{3-x}(OH)_y$ (denoted as S2) was successfully synthesized via a solid-state synthetic route by utilizing $In_2O_3$ nanocrystals (denote as S1) and $NaBH_4$ as precursor materials. The pale yellow photothermally inactive form of $In_2O_3$ nanocrystals was transformed into black photothermally active $H_zIn_2O_{3-x}(OH)_y$ nanocrystals containing surface frustrated Lewis pairs (SFLPs) as active sites. The new black $H_zIn_2O_{3-x}(OH)_y$ enabled a tandem methanol synthesis process via an initial RWGS reaction followed by a CO hydrogenation reaction, where by-product CO functions as in-situ feedstock for methanol synthesis.

## Results

**Structural characterization.** As shown in Supplementary Fig. 1, $In_2O_3$ nanoparticles and $NaBH_4$ (weight ratio is 1:1.5) were added in an agate mortar, ground for 15 min, and then transferred to a small crucible. They were then calcined in a muffle furnace at 350 °C for 30 min. Finally, the obtained powder was washed with water several times and dried in a vacuum oven to obtain S2. Scanning electron microscopy (SEM) images of S2 were obtained and indicated a similar appearance and slightly larger particle size compared to the parent $In_2O_3$ nanoparticles (Supplementary Fig. 2). The specific surface areas for S1 and S2 were 51 and 57 $m^2$ $g^{-1}$, respectively (Supplementary Fig. 3). According to the UV–Vis-NIR in Supplementary Fig. 4, the as-prepared S2 exhibited stronger absorbance than the S1, which could be caused by higher concentration of surface [O]. Due to the strong phonon confinement induced by the defects, the Raman spectrum of S2 exhibits a redshift, and the broad peaks of S2 represent the amorphization, Supplementary Fig. 5[6,17].

The similar powder X-ray diffraction (XRD) patterns of S1 and S2 indicate no major structural change in their cubic bixbyite structure (Fig. 1a). The absence of Na and metallic indium in the PXRD pattern of S2 is consistent with the inductively coupled plasma mass spectrometry (ICP-MS) and X-ray photoelectron spectroscopy (XPS) results. Similarly, in Fig. 1b, the high-resolution transmission electron microscopy (HRTEM) images, the (222) lattice spacing of S2 is 0.292 nm (Supplementary Fig. 6), which agrees well with the XRD results. As shown in Fig. 1c and

Supplementary Tables 1, S1 contains about 24.23% surface [O], implying [O] was the major active site for the reaction. In sharp contrast, S2 contains 44.87% of [O] and 11.25% of the OH group, which confirms the presence of the possible dual active sites, [O] and the SFLPs[26]. In Supplementary Fig. 7a, the $In3d$ valence spectra of S1 and S2 show no metallic indium, which agrees well with the PXRD result and demonstrates the blackish color is not caused by the presence of metallic indium. For $H_2$ temperature-programmed desorption ($H_2$-TPD), the three distinct absorption peaks for S2 could correspond to the physical adsorption of $H_2$ on the sample surface at lower temperatures (<150 °C), desorption of surface indium hydrides (200–300 °C), and surface protonated hydroxides (>300 °C) at higher temperatures (Supplementary Fig. 8a), respectively[27,28]. The solid-state H-NMR spectrum of S2 confirms the presence of hydride species, which could potentially enhance the absorption of $CO_2$ and CO species (Supplementary Fig. 8b)[26,29].

Furthermore, the in situ X-ray absorption near edge structure (XANES) spectrum for S2 was conducted in various atmospheres and temperatures and confirmed the absence of metallic indium, Supplementary Fig. 9. The reference spectrum was obtained at room temperature in the air. It can be seen from Fig. 1d that the S2 spectra shifts to higher excitation energy under $N_2$ and reaction atmosphere at 260 °C, which indicates the slightly increased oxidation state of indium. The increase in oxidation state could be caused by loss of surface hydrides at higher temperatures, whereas under the absence of $H_2$ ($N_2$ atmosphere) conditions, the excitation energy shifts the most, and the one with $H_2$ (dark and light) shifts the least and could regenerate the hydrides on the S2 surface.

To explore the detailed structural information of S2 during the reaction, in situ X-ray absorption spectroscopy (XAS) at different temperatures and 50% $H_2$ and $CO_2$ was conducted (Supplementary Figs. 10, 11). While increasing the temperature from 200 to 300 °C without light, the coordination numbers of indium increased with a slightly shorter bond length with respect to the one at 200 °C (Supplementary Fig. 11a). This could have been caused by the enhanced $H_2$ splitting ability, which decreased the average bond length with increasing coordination numbers (Supplementary Fig. 11b). Interestingly, when irradiated with light, the coordination number of indium eventually decreased and exhibited the opposite trend with respect to the dark condition. Although the $H_2$ splitting ability could be enhanced at higher temperatures (200 and 260 °C), the desorption of the $H_2$ molecule and formation of [O] could be induced by light irradiation, resulting in a lower coordination number. The prolonged bond length of indium may also indicates the formation of [O].

**Catalytic performance.** The photothermal catalytic performance of CZA, S1, and S2 without an external thermal source is shown in Fig. 2a, with the original GC spectrum shown in Supplementary Fig. 12. Owing to the significant difference between the absorption abilities of the solar spectra of S1 and S2 (Supplementary Fig. 4), S2 exhibits remarkable photothermal catalytic performance, while S1 shows negligible activity. By contrast, CO is the only product from CZA catalyst under the same condition. For the S2 sample, despite the prominent CO generation from the RWGS reaction, methanol can also be detected as a product at the rate of 31.2 μmol $g^{-1}$ $h^{-1}$ with a selectivity of 36.7%. To the best of our knowledge, due to the exothermic nature of methanol synthesis, it is yet to be achieved via photothermal $CO_2$ hydrogenation without any external thermal source[19,30–32]. The methanol selectivities over samples in a batch reactor and a flow reactor were shown in Fig. 2b. The data of thermodynamic equilibrium under similar conditions were calculated (Supplementary Fig. 13). The results

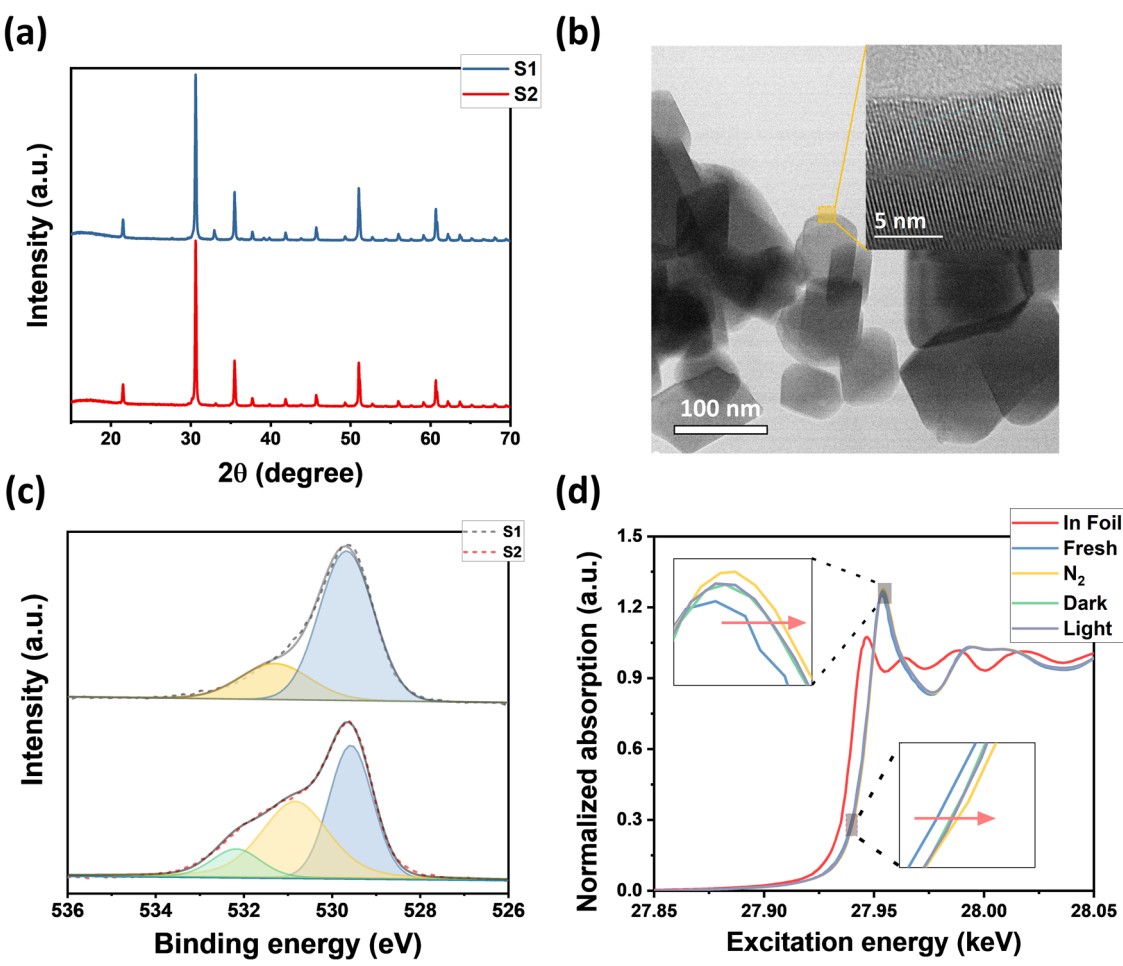

**Fig. 1 Structural characterizations. a** Powder XRD patterns of S1 and S2. **b** TEM image of S2 and high-resolution TEM image inserted in the upper right. **c** High-resolution O1s core level XPS spectrum of S1 and S2 (the dashed line is the original XPS spectra, and the solid line is the best fit results). **d** In situ XANES of S2 catalyst under different conditions.

showed that the methanol selectivity could only achieve 0.04% with 50% $H_2$ and 0.1% with 75% $H_2$ at 250 °C and atmospheric pressure, which confirms that methanol synthesis is challenging under atmospheric pressure.

To further study the detailed reaction mechanism of S2, the sample was tested in a flow reactor under various conditions. At atmospheric pressure and low $H_2$ concentration (50%), both CO rate and methanol rate increased with temperature and exhibited photo-enhancement when irradiated with light (Fig. 2c). Such a phenomenon is mainly caused by the endothermic nature of the RWGS reaction, due to which thermal energy can significantly enhance the reaction rate. The CO rates at 200 °C were 0.49 and 0.98 $\mu mol\ g^{-1}\ h^{-1}$ with and without irradiation, respectively, exhibiting a photo-enhancement of about 100%. When the temperature increased to 300 °C, the CO rates were 226.79 and 433.68 $\mu mol\ g^{-1}\ h^{-1}$ for dark and light, respectively, with more than 90% photo-enhancement. The corresponding Arrhenius plot indicates two almost parallel plots for dark and light with activation energies of 138.43 and 141.43 $kJ\ mol^{-1}$, respectively (Fig. 2d). The similar activation energies imply a similar reaction mechanism and solar energy being mainly converted into thermal energy to enhance catalytic performance. The corresponding photothermal advantage can be estimated as about 16.4 °C by fitting the light CO parameters into the dark Arrhenius plot. The resulting photothermal advantage for S2 accounts for 91.2% enhancement (from 226.79 to 433.68 $\mu mol\ g^{-1}\ h^{-1}$ at 300 °C. The calculation details are shown in Supplementary Information).

More interestingly, due to its strong tendency for methanol synthesis, S2 can produce methanol even with 50% $H_2$ concentration ($CO_2$:$H_2$ = 1:1 in Fig. 2e; the methanol selectivity is shown in Supplementary Fig. 14). The methanol rate for dark increased from 2.05 $\mu mol\ g^{-1}\ h^{-1}$ at 250 °C to 18.95 $\mu mol\ g^{-1}\ h^{-1}$ at 300 °C, and from 1.49 $\mu mol\ g^{-1}\ h^{-1}$ at 230 °C to 23.03 $\mu mol\ g^{-1}\ h^{-1}$ at 300 °C with light. The temperature difference for the starting point of methanol synthesis implies similar photothermal enhancement for the RWGS reaction. Unlike in previous studies, the methanol rate is inhibited by high temperatures with a notable loss[1,2,30]. The methanol rate of S2 reached a plateau at the temperatures of 290 and 300 °C, which suggests a unique reaction mechanism. Therefore, due to the thermal plateau of methanol rate, the photo-enhancement of the methanol rate could be considered as the photochemical contribution, which accounts for 21.3% enhancement (from 18.98 to 23.03 $\mu mol\ g^{-1}\ h^{-1}$ at 300 °C). To further confirm the possibility of a photochemistry-enabled reaction pathway, temperature-programmed photoluminescence (PL) was conducted. S2 has a lower PL intensity than S1, indicating a lower rate of radiative recombination. Due to the higher concentration of [O] and the lower rate of radiative recombination, the recombination efficiency of photo-excited electron-hole pairs over S2 could be lower than that of S1[33–35] (Supplementary Fig. 15). The isotope label test was also conducted to further confirm that the produced methanol is coming from the $CO_2$, and the presence of $^{13}C$ methanol is shown in Supplementary Fig. 16.

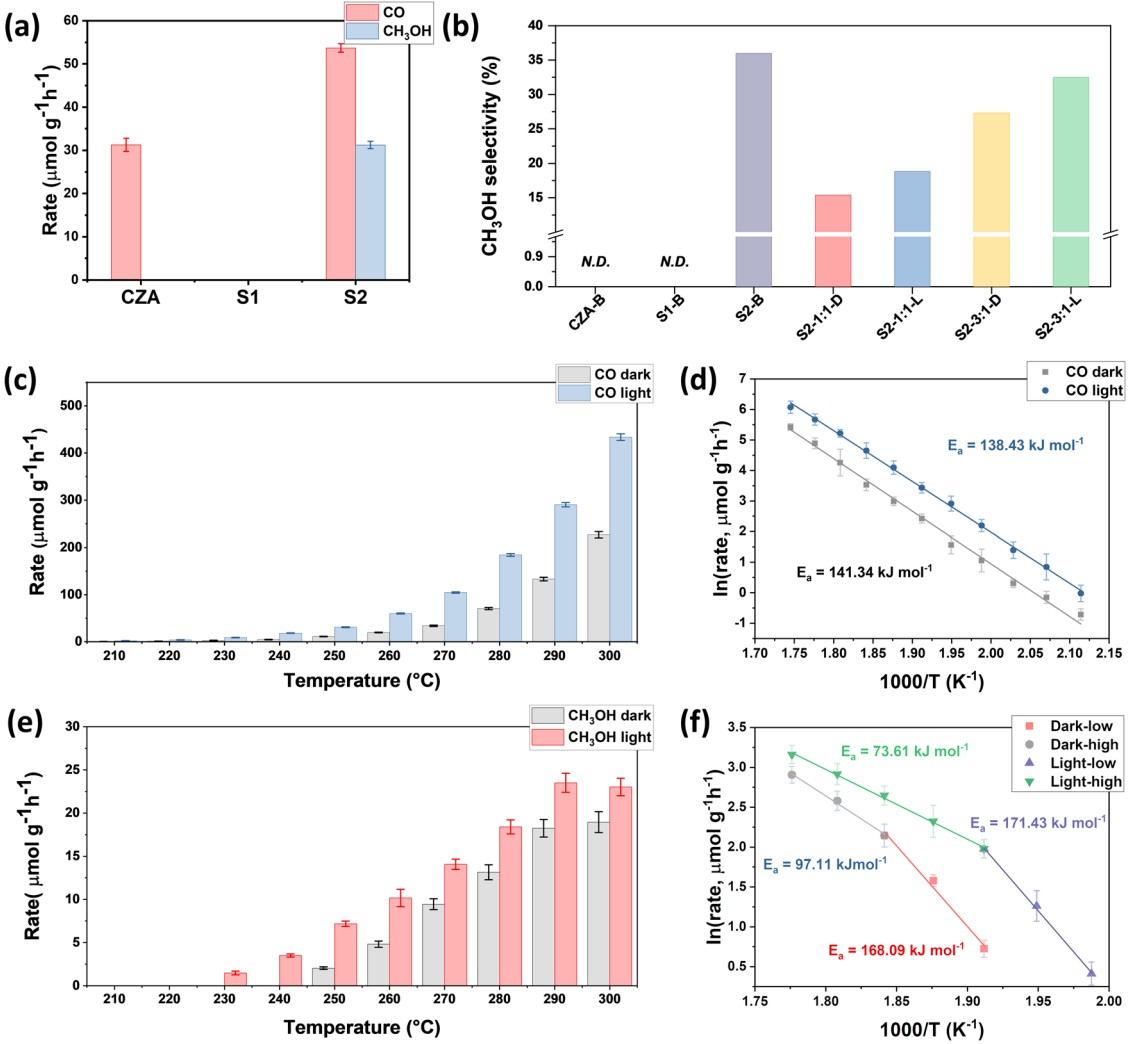

**Fig. 2 Catalytic performance of the samples. a** Photothermal catalytic performance of Cu/ZnO/Al$_2$O$_3$(CZA), S1, and S2 in the batch reactor. Reaction conditions: H$_2$/CO$_2$ ratio = 3:1, 30 Psia, and ~20 suns light intensity with a duration of 0.5 h without external heat. **b** Methanol selectivity with different H$_2$: CO$_2$ ratios; D indicates dark, L indicates light, and B indicates batch reactor. **c** CO rate of S2 in a flow reactor under light/dark conditions, and **d** the corresponding Arrhenius plot. **e** Methanol rate of S2, and **f** the corresponding Arrhenius plot. Conditions for flow measurement: atmospheric pressure, H$_2$/ CO$_2$ ratio = 1:1 with a total flow rate of 4 mL min$^{-1}$, and light intensity of 6 suns.

The corresponding methanol Arrhenius plots are very different from the CO Arrhenius plots (Fig. 2f). Two-stage convex Arrhenius plots were obtained for both dark and light conditions. Similar to the CO plot, at lower temperatures, the plots exhibited similar activation energies at 168.09 and 171.43 kJ mol$^{-1}$ for dark and light, respectively, which implies a very similar reaction mechanism. To further confirm the reliability of this method, the methanol rate with light was fitted into the dark Arrhenius plot, and it provided a photothermal effect of about 16.9 °C, which agrees well with the estimated temperature from CO plots. Such a phenomenon also reveals that at low temperatures (<250 °C for light and <270 °C for dark), the photothermal effect dominates the photo-enhancement of both the RWGS reaction and methanol synthesis. In sharp contrast, at higher temperatures, the second stage of dark and light plots initiated very different activation energies of 97.11 kJ mol$^{-1}$ without light and 73.61 kJ mol$^{-1}$ with light, indicating a different catalytic mechanism and involved the participation of photochemistry in the reaction. Under the dark condition, the activation energy of methanol synthesis decreased from 168.09 to 97.11 kJ mol$^{-1}$. The typical convex Arrhenius plot indicates different kinetic controlled pathways[36,37], whereas at lower temperatures, the reaction mainly

depends on the activation and the dissociation of CO$_2$ molecules, and at higher temperatures, it mainly depends on the diffusion of reaction intermediates such as *CO[36,38,39]. When light was shone onto S2, the estimated activation energy further decreased to 73.61 kJ mol$^{-1}$, implying the light irradiation could benefit the reaction intermediate diffusion process (*CO could be transferred from the active SFLPs sites to the [O] sites) and improve the methanol rate.

To further study the stability of S2, a 75 h continuous stability test was conducted in a flow reactor. To achieve the best methanol rate and selectivity, the catalytic performance of S2 was measured with 75% H$_2$ (H$_2$:CO$_2$ = 3:1) at 250 °C, resulting in an initial methanol rate of 14.92 μmol g$^{-1}$ h$^{-1}$ with a selectivity of 32.6% (Fig. 3a). However, as time passed, the CO rate dropped from 30.89 to 7.77 μmol g$^{-1}$ h$^{-1}$ (a 75% drop), and the methanol rate dropped to 7.54 μmol g$^{-1}$ h$^{-1}$ (50% drop) with a final methanol selectivity of 49.23%. Although the methanol selectivity significantly improved, the methanol rate was inhibited, which could have been caused by the production of water, Eqs. 1 and 2[2,10,40].

Therefore, another stability test under 50% H$_2$ at 250 °C was conducted for S2, which resulted in very different catalytic

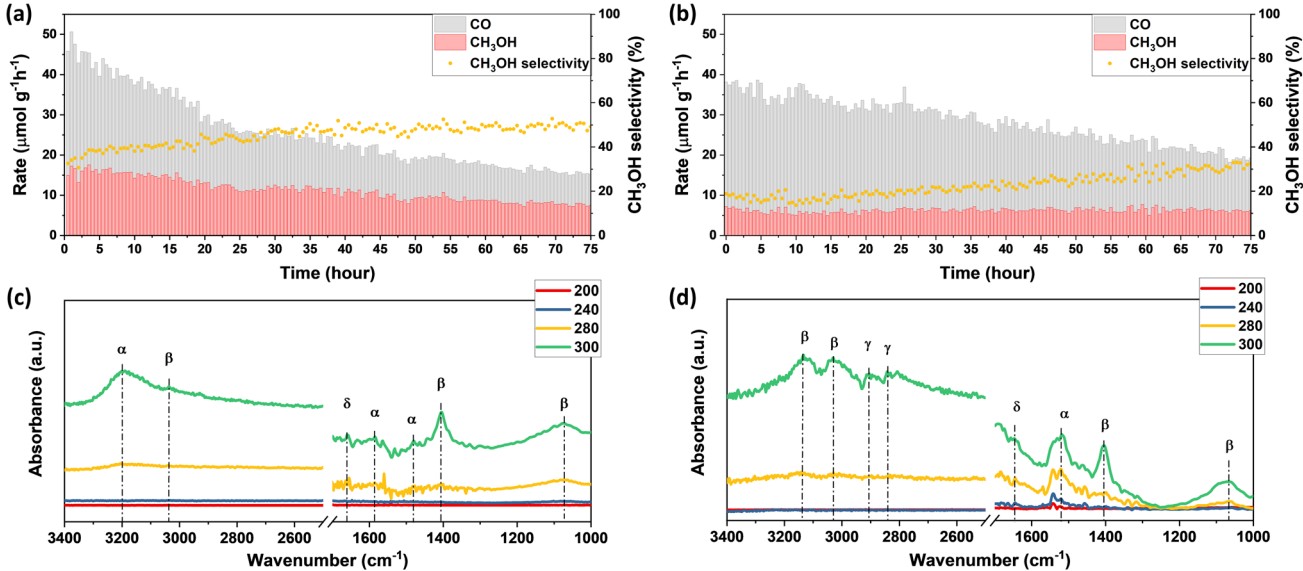

**Fig. 3 Stability test and in-situ DRIFTS analysis.** Seventy-five hours continuous stability test for S2 at 250 °C with light under **a** 75% $H_2$ and **b** 50% $H_2$. Reaction conditions: atmospheric pressure, $H_2$/$CO_2$ ratio of 1:1, a flow rate of 4 mL min$^{-1}$, and light intensity of ~6 suns. **c** In situ DRIFTS of S2 with $CO_2$ and $H_2$ (1:1) and **d** with CO and $H_2$ (1:1). All DRIFTS spectra were subtracted by the background signal of S2 obtained under He. α: C=O stretching vibrations (formate); β: C–H and C–O stretching vibrations (methoxy); γ: C–H stretching vibrations (methanol); δ: H–O stretching vibrations ($H_2O$).

performance (Fig. 3b). The initial CO rate was 30.98 μmol g$^{-1}$ h$^{-1}$, and the initial methanol rate was 7.2 μmol g$^{-1}$ h$^{-1}$ with a methanol selectivity of 18.9%. Similarly, as time passed, the CO rate was inhibited significantly and stabilized at 13.01 μmol g$^{-1}$ h$^{-1}$ (42% drop), while 90% of the methanol rate was still preserved (6.48 μmol g$^{-1}$ h$^{-1}$) with a methanol selectivity of 33.2%. The different deactivation processes indicate two distinct catalytic centers for the RWGS reaction and methanol synthesis, possibly SFLPs and [O]. The water produced by the RWGS reaction could have caused strong inhibition of CO rate. The negligible inhibition of methanol synthesis could have been caused by the subsequent CO hydrogenation which did not produce additional water molecules, Eq. 3. To further confirm that CO can be used as the feedstock for methanol synthesis, another CO hydrogenation test was conducted with CO:$H_2$ = 1:1 (Supplementary Fig. 17). The methanol signal could be observed from 210 to 300 °C over S2 and barely from S1. This finding confirms that the as-prepared black indium oxide could use CO as the reactant for methanol synthesis.

In Supplementary Figure 18, the SEM image and TEM image of the posted S2 show unchanged particle size and morphology, and the lattice space is about 0.292 nm corresponding to $In_2O_3$ (222). The crystalline structure of the used S2 is the same as that of the fresh S2, shown in Supplementary Fig. 19. Furthermore, the used S2 O1s spectra also exhibit a similar composition of the fresh S2 (Supplementary Fig. 7b), indicating excellent stability.

In situ DRIFTS was used to track the intermediate species to understand the mechanism of methanol formation on S2 (Fig. 3c, d, and Supplementary Fig. 20). The ratio of $H_2$:$CO_2$ was 1:1, and the peak at 1070 cm$^{-1}$ represents methoxy species of C–O stretching vibrations[31,41]. The observed methoxy species indicate the formation of methanol. Meanwhile, the formate could either be the source of methanol or CO. To further confirm the methanol pathway and the possibility for tandem methanol synthesis, a 50% $H_2$ and CO in situ DRIFTS measurement was conducted, Fig. 3d. Despite the presence of methanol-related peaks at 2800–2900 cm$^{-1}$ of C–H stretching[42,43] and methoxy species of C–H stretching vibrations at 3000–3100 cm$^{-1}$[19], the absence of typical formate species at 3200 cm$^{-1}$ of C–H stretching and 1580 cm$^{-1}$ of OCO asymmetric stretching vibrations implies

that the formate species could be the intermediate species for RWGS rather than for methanol synthesis. It is worth noting that the peak at 1520 cm$^{-1}$ could correspond to the asymmetric HOCO* species stretching vibrations[31]. Since the DRIFTS was conducted under CO and $H_2$ atmosphere, the HOCO* was not supposed to be form unless the In–OH group assisted SFLPs addition reaction was conducted. Moreover, the CO temperature-programmed desorption (CO-TPD) shown in Supplementary Fig. 21 indicates strong CO adsorption on the S2 surface. This strong CO absorption could further benefit the subsequent CO hydrogenation in tandem methanol synthesis.

## Discussion

Spin-polarized density functional theory (DFT) simulations were performed over a 1*1 (110) indium oxide surface to verify the proposed tandem methanol synthesis pathway (Fig. 4 and Supplementary Fig. 22). Oxygen vacancies [O] and an end-on hydroxyl group (OH) on the indium atom, which is vicinal to [O], were crafted to form the experimentally identified SFLP sites (InOH•••In, Supplementary Fig. 23). These SFLPs heterolytically dissociated the $H_2$ molecule to form H$^+$ (Mulliken electronegativity: 0.46 a.u.) and H$^-$ (Mulliken electronegativity: −0.36 a.u.) in the form of InHOH$^+$•••InH$^-$ ($E$ = −0.93 eV), where bond lengths for O–H and In–H were 0.979 and 1.795 Å, respectively (Fig. 4a). The differential charge density map also evidenced electron density accumulation for the In–H bond and depletion from the hydroxyl group.

Upon $CO_2$ adsorption on the protonated and hydridic SFLP sites, the endothermic RWGS reaction occurred on the surface. Both carbonate and formate were possible intermediates, and the in situ DRIFTS experimentally identified the latter. Desorption of In-bonded CO to regenerate catalytic SFLP sites should overcome an energy barrier of 0.84 eV, which would be consistent with the high-temperature desorption peaks in the CO-TPD spectra in Supplementary Fig. 21.

Alternatively, the In-bonded CO could be readily used as the feedstock for further methanol synthesis to avoid direct desorption (Supplementary Fig. 24). An exothermic conformation

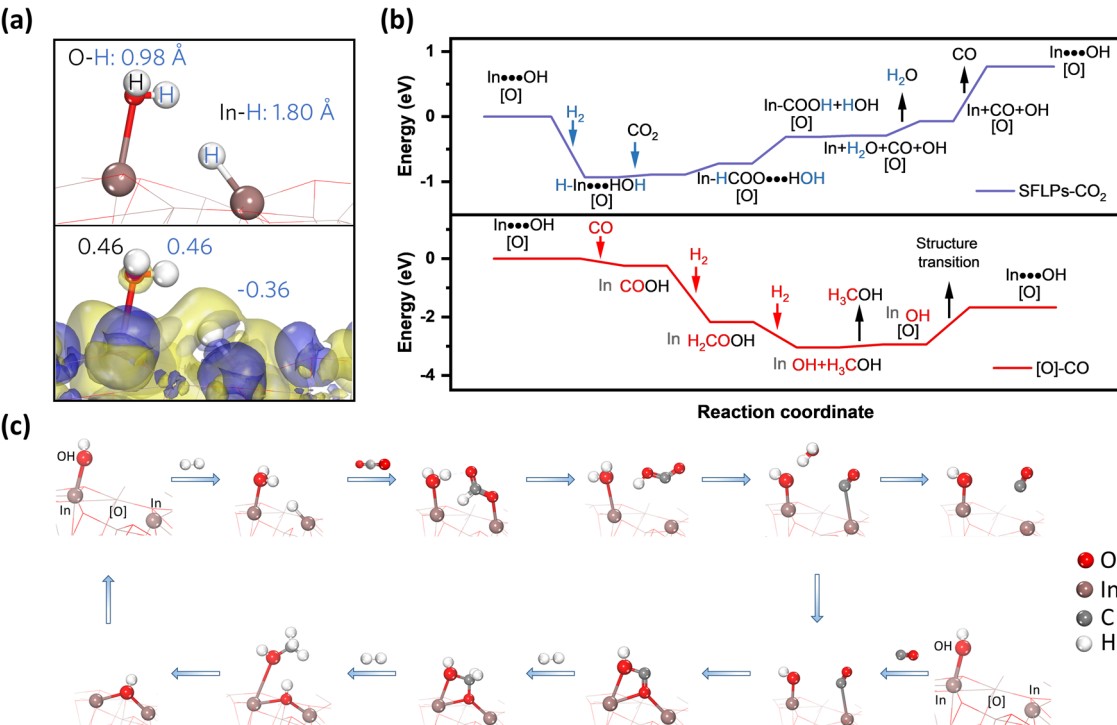

**Fig. 4 DFT calculation for mechanism analysis. a** Mulliken electronegativity analysis (Up) and corresponding differential charge density of protonated and hydridic SFLP sites (Down). **b** Free energy diagram of the RWGS reaction over SFLP sites and subsequent methanol synthesis over the [O] site. **c** The proposed tandem reaction mechanism for methanol synthesis and corresponding atomic configurations.

transition was observed when the CO (HO-In-[O]-In-CO) was trapped by [O] via an end-on mode ($E = -0.17$ eV). The nucleophilic oxygen atom refilled the [O] and the electrophilic carbon atom bonded to the OH, forming a bridging two-coordinated $sp$ carbon (OCOH) species that was prone to be hydrogenated to $sp^3$ hybridization. The resulting product was an OCH$_2$OH species. Further hydrogenation would break the C–O bond intrinsic to the CO molecule, refilling the [O] with a hydroxyl group and forming an O-terminated methanol with its methyl group repelled by the surface hydroxyl group. Such an end-on absorbed methanol was highly favorable for desorption ($E = 0.20$ eV). After the release of CH$_3$OH molecule, the [O] of the catalyst regenerated as the [O]-confined bridging OH underwent an endothermic structure transition to terminal OH with $E = 1.27$ eV. Notably, this energy demand could be fully compensated by the heat release from previous H$_2$ adsorption and activation steps ($E = -1.93$ and $-0.87$ eV). Based on the low energy barriers of SFLP-related methanol synthesis, one can conclude that the methanol synthesis would be more favorable once the reaction temperature is high enough to ensure adequate CO production and SFLP recovery. This is in good agreement with the high temperature-related low apparent $E_a$ for methanol synthesis observed in Fig. 2f.

It is worth noting that direct CO$_2$ hydrogenation to methanol on the [O] of In$_2$O$_3$ is also feasible, which has been studied by Ge et al.[44] and Sun et al.[45] recently. The pathway is believed to be "CO$_2$-H$_x$CO$_2$-CH$_3$OH" ($x = 1$, 2, and 3) with corresponding activation energy barrier (0.64–2.52 eV) associated with the polymorph, exposed facet of In$_2$O$_3$ and the position of the surface oxygen vacancy. The major difference between our SFLPs model and the [O] model is the involvement of an [O]-vicinal hydroxyl group. To understand this Lewis base OH-induced difference during CO$_2$-to-methanol conversion, we also calculated the free energy diagram on the OH-free [O] sites of the In$_2$O$_3$ (110) facet, which was constructed on the top of the SFLP model by

abstracting the OH group. Distinct to the "downhill" methanol assembly in the SFLP-related pathway, energy barriers of 2.99 eV via an In-associated stepwise hydrogenation or 1.53 eV via a combined atomic and molecular hydrogenation pathway were observed (Supplementary Fig. 25). This result agreed with previous computational work, indicating our SFLP-associated tandem RWGS-methanol synthesis pathway could be the possible reason for the ultra-high methanol selectivity.

In summary, a new black form of a photothermally active indium oxide catalyst was successfully prepared from a photothermally inactive indium oxide via a solid-state synthesis method with a methanol selectivity of 30–50% at ambient pressure. The pathway to methanol was explored experimentally and theoretically and revealed the operation of a tandem reaction scheme in which by-product CO from RWGS acted as an in-situ feedstock for the formation of methanol. The tandem process shifts the conventional competing RWGS and methanol synthesis process into a combined reaction pathway in the flow reactor system. Through surface site engineering, the new black indium oxide photothermal catalyst overcomes the thermodynamic constraints that control the conventional synthesis of methanol. The observed boost in selectivity under atmospheric pressure conditions bodes well for the development of a solar refinery for the production of sustainable methanol.

## Methods
### Catalysts preparation
*Chemicals.* Commercial In$_2$O$_3$ nanocrystal was purchased from Alfa Aesar.

*Synthesis of the black indium oxide.* Commercially available In$_2$O$_3$ and NaBH$_4$ (weight ratio 1:1.5) were added into an agate mortar and ground for 15 min. The mixture was then transferred into a muffle furnace to be calcined at 350 °C for 30 min (In$_2$O$_3$ + NaBH$_4 \rightarrow$ In$_2$O$_{3-a}$H$_b$). After cooling to room temperature, a large amount of deionized water was used to rinse excess NaBH$_4$ (In$_2$O$_{3-a}$H$_b$ + H$_2$O $\rightarrow$ H$_z$In$_2$O$_{3-x}$(OH)$_y$ + δH$_2$). Finally, the black In$_2$O$_3$ was obtained in an oven at 100 °C overnight.

*Synthesis of the Cu/ZnO/Al₂O₃ (CZA)*. CZA catalyst is synthesized from a zinc malachite precursor (Cu,Zn)₂(OH)₂CO₃ (Cu:Zn = 70:30) with additional 13 mol% Al (metal basis) by co-precipitation[46–48]. Co-precipitation was performed using sodium carbonate solution as the precipitating agent at pH of 6.5 and $T = 65\,°C$. The precipitate was aged in the mother liquor at 65 °C for 30 min. The precursors were obtained by washing with water and ethanol several times, and then placed in an oven to dry at 60 °C. Finally, the precursors were calcined in static air at 330 °C (2 °C/min) in a muffle furnace.

**Catalyst characterizations**. The crystal structure of the catalysts was performed using powder XRD. XRD analysis was conducted on a Bruker D8 Advance with Cu Kα radiation source (λ = 1.5406 Å). The hydrogen temperature-programmed desorption (H₂-TPD), CO temperature-programmed desorption (CO-TPD), and CO₂ temperature-programmed desorption (CO₂-TPD) experiments were performed on a Micromeritics AutoChem 2920 instrument. XPS was carried out using a VG Thermo ESCALAB 250 spectrometer with an Al K_α X-ray source operating at 15 kV and 27 A. The binding energy of samples was calibrated using the carbon C 1s at 284.6 eV. SEM was performed on Hitachi SU5000 Schottky field emission SEM. The high-resolution TEM images were performed by Hitachi HF-3300 cold field emission TEM at an accelerating voltage of 300 kV. The sample d-spacing and lattice plane were analyzed using the DigitalMicrograph. Nitrogen Brunauer-Emmett-Teller (BET) adsorption isotherms were obtained at 77 K using a Micromeritics ASAP 2010 instrument. The samples' diffuse reflectance was measured using a Lambda 1050 UV-Vis-NIR spectrometer from PerkinElmer and an integrating sphere with a diameter of 150 mm. Fluorescence spectroscopy (PL, Hitachi F-4500, Japan) was applied to study the separation of photo-induced carriers. The samples were excited by a 325 nm laser light.

The ¹H solid-state MAS NMR spectra were obtained at a spinning rate of 12 kHz. The NMR Spectra were calibrated to reference adamantane with optimized parameters: pulse width (pwX90) = 3.45 microseconds, fine power (aX90) = 2700, course power (tpwr) = 59, and synthesizer offset (tof) = 1192.9. Number of scans = 64 and delay time = 6 s. Samples for ¹H solid-state MAS NMR were treated by H₂ and then transferred to a glove box with an Ar atmosphere for sample loading[22].

In situ diffuse reflectance infrared Fourier-transform (DRIFTS) spectra were carried out on Nicolet iS50 spectrometer with a liquid-nitrogen-cooled MCT detector. Firstly, the samples were pretreated at 200 °C for 1 h in He at a 20 mL min⁻¹ flow rate and then cooled down to room temperature. Subsequently, a background spectrum was collected in He before the catalyst exposure to analysis gas. The catalysts would be exposed to mixed gas (mixed gas 1:10% CO, 10% H₂ 80% Ar; mixed gas 2: 10% CO₂, 10% H₂ 80% Ar) at a flow rate of 20 mL min⁻¹.

XANES and extended X-ray absorption fine structure (EXAFS) were measured, at beamline 20BM of the Advanced Photon Source at Argonne National Laboratory. A Si (111) double-crystal monochromator and the focused beam was used to perform the measurements at In K-edge (27940 eV). Harmonic rejection was facilitated by detuning 30% of the beam intensity at 1000 eV above the edge of interest. Data were collected in transmission mode using 20% Ar and 80% N₂ for all ionization chambers. Details on the beamline optics and instruments can be found elsewhere[49].

The sample was measured inside a quartz capillary inserted in a cell where the temperature could be raised under flowing gases conditions. In Supplementary Fig. 9a, a LED was placed on the side of the cell to illuminate the sample and the output spectrum of LED lamp was given in Supplementary Fig. 9b. At room temperature, the catalyst was exposed to the reaction condition 5%H₂/N₂ 20 mL min⁻¹ and 10%CO₂/He 10 mL min⁻¹ and the temperature increased to 200 °C. At 200 °C, EXAFS measurements were performed in dark and then under light conditions. The light was turned off and the temperature increased to 260 °C. Again EXAFS measurements were performed at dark and light conditions. The same procedure was repeated increasing the temperature to 300 °C. At the end, the sample was cooled down to room temperature and the CO₂:H₂ ratio changed to 1:2. The temperature and light procedure was repeated. Afterward once more the procedure was repeated but using CO₂:H₂ ratio 1:3.

The methanol equilibrium selectivities were estimated by ASPEN. The simulation for calculating the methanol equilibrium yields and selectivities used the Equilibrium reactor block in Aspen Plus V11 with the ideal NRTL property package. The specified reactions were RWGS and methanol synthesis. The feed used was 0.67:0.33 H₂:CO₂ kmol/h at 250 C at different conditions.

**Catalytic activity measurements**. The gas-phase photothermal catalytic evaluations were conducted in a custom-built 1.5 mL stainless steel batch reactor with a fused silica viewport sealed with Viton O-rings. The reactor was evacuated using an Alcatel dry pump prior to being purged with the reactant H₂ gas (99.9995%). After purging the reactor, it was filled with a 3:1 stoichiometric mixture of H₂ (99.9995%) and CO₂ (99.999%) until the total pressure reached 30 Psia. Reactors were irradiated with a light intensity of 20 suns for a duration of 0.5 h without external heat.

Gas-phase flow reactor measurements were carried out in a fixed-bed tubular reactor (3 mm outer diameter and 2 mm inner diameter). During the reaction, H₂ (Praxair 99.999%) and CO₂ (Praxair 99.999%) were flowed in a 1:1 ratio at a total volumetric flow rate of 4 ml/min. For photocatalytic rate measurements, the reactor was irradiated with a 300 W Xe lamp (PLS-SXE300D, Beijing Perfectlight). Product gases were analyzed using FID and TCD installed in an SRI-8610 gas chromatograph equipped with a 3 in. Mole Sieve 13a and a 6 in. Haysep D column.

**DFT calculation**. All spin-polarized calculations were performed using the CASTEP package[50] with the following convergence criteria: energy 5.0e−5eV/atom, max. force 0.1 eV/Å, max. stress 0.2 GPa, max. displacement 0.005 Å, and SCF tolerance 1.0e−5 eV/atom. According to previous theoretic work, the (110) surface was chosen as the reaction surface, which herein consisted of 1*1 In–O layers of 8.02 Å in thickness and a vacuum slab of more than 14 Å. The oxygen vacancy ([O])-laden surface was crafted by abstracting a bridging O atom between two surface-In atoms, where a terminal hydroxyl group was then added to form the surface-FLPs-laden surface. The GGA-RPBE functional was used for the exchange-correlation potential, and the plane-wave pseudopotential approach and ultrasoft pseudopotentials were employed for all the atoms with a kinetic energy cutoff of 520 eV. All models were first fully relaxed via geometry optimization and then applied for the energy calculation. The charge density difference was calculated using set of atoms, and the blue and yellow isosurfaces represented electron density accumulation and depletion, respectively, where the absolute isovalue was 0.01. The atomic Mulliken electronegativity was calculated using population analysis.

The adsorption energies of adsorbates are defined as[51]

$$E_{ad}(m) = E_{m-s} - E_s - E_m$$

where $m$ represents molecular adsorbate and $s$ represents the surface of indium oxide.

## Data availability

All data are available in the main text or the Supplementary Information. Source data are provided with this paper.

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

## Acknowledgements

All authors appreciate the support of the National Key R&D Program of China (Grants No. 2021YFF0502000), National Natural Science Foundation of China (Grants No. 52102311), The Program for Guangdong Introducing Innovative and Entrepreneurial Teams (Grants No. 2019ZT08L101), The Special Fund for the Sci-tech Innovation Strategy of Guangdong Province (Grants No. 210629095860472), The Shenzhen Natural Science Foundation (Grants No. GXWD20201231105722002-20200824163747001), The Shenzhen Key Laboratory of Eco-materials and Renewable Energy (Grants No. ZDSYS20200922160400001), and The University Development Fund (Grants No. UDF01001721). G.A.O. acknowledges the financial support of the Ontario Ministry of Research and Innovation (MRI), the Ministry of Economic Development, Employment and Infrastructure (MEDI), the Ministry of the Environment and Climate Change's (MOECC) Best in Science (BIS) Award, Ontario Center of Excellence Solutions 2030 Challenge Fund, Ministry of Research Innovation and Science (MRIS) Low Carbon Innovation Fund (LCIF), Imperial Oil, the University of Toronto's Connaught Innovation Fund (CIF), Connaught Global Challenge (CGC) Fund, and the Natural Sciences and Engineering Research Council of Canada (NSERC). This research used resources of the Advanced Photon Source, an Office of Science User Facility operated for the U.S. Department of Energy (DOE) Office of Science by Argonne National Laboratory, and it was supported by the U.S. DOE, under Contract No. DE-AC02-06CH11357, and the Canadian Light Source and its funding partners. C.M. acknowledges the financial support from University of Toronto's Arts & Science Postdoctoral Fellowship and the National Supercomputer Center in Shenzhen (China) for providing high-performance computation. Z.Z. appreciates the support of the China Postdoctoral Science Foundation (Grants No. 2021M703074).

## Author contributions

L.W., Z.Z., and G.A.O. conceived and designed the experiments. L.W., J.S., C.M., and Z.Z. synthesized the materials and performed in situ DRIFTS and catalytic measurements. C.M. conducted the DFT simulation. A.A.T. performed ASPEN simulations. D.M.M. and P.N.D. performed and analyzed the in situ XAS. C.Q. and J.H. conducted and analyzed the STEM and HRTEM. R.S., S.T., Z.L., J.S., Z.Z., X.D., J.Y., and W.T. conducted BET, XPS, PL, and other basic characterizations and catalytic performance. L.W., G.A.O., Z.Z., and C.M. wrote the manuscript. All authors discussed the results and commented on the manuscript.

## Competing interests

The authors declare no competing interests.
