## [Peer Review File · Nature Communications]

Title: New Black Indium Oxide – Tandem Photothermal CO₂-H₂ Methanol Selective CatalystREVIEWER COMMENTS

Reviewer #1 (Remarks to the Author):

The authors report the black indium oxide catalyst for photothermal CO₂ hydrogenation into CH₃OH at a low hydrogen concentration and ambient pressure. Although they employ both the experimental and theoretical methods to verify this way, the selectivity of CH₃OH is too low and the overall property is unattractive. Besides, they seemingly ignore the discussion about the effect of light relative to the heat for this process. Therefore, the quality of this work does not meet the high standard for "Nat. Commun.". Moreover, there are several other issues that dampen my enthusiasm for this manuscript:

1. Fig. S2 show the SEM images and size distributions of S1 and S2. There is a good deal of particles in the SEM images and how their size distributions are determined?
2. Where are the original XPS spectra for S1 and S2 in Fig. 1C? Whether the XPS spectra in the manuscript match with their original XPS spectra?
3. Can the positions of the valence-band maximum and conduction-band minimum meet the theoretical potential of CO₂/CH₃OH?
4. What is the opposite half reaction relative to the CO₂ hydrogenation into CH₃OH?
5. Fig. S11 shows the chromatography spectra signal of CH₃OH. Is it a gas form or liquid form?
6. The isotope-labeled mass spectrum in Fig. S15 is incomplete. Are there the peaks of other smaller species (¹³CH₃, ¹³CH₂, ¹³CH) and ¹³CO₂?
7. How the properties of photocatalytic CO₂ reduction by these two catalysts? The comparative experiments should be added.
8. The authors should compare the property of CO₂ hydrogenation in this work with other reported references (10.1038/s41565-018-0089-z; 10.1126/sciadv.aaz2060).

Reviewer #2 (Remarks to the Author):

The manuscript by Zhang et al. describes the a modified In₂O₃ catalyst that is capable of synthesizing methanol from CO₂ (or CO) and H₂. The modified In₂O₃ is black and demonstrates photothermal activity under Xe lamp irradiation. The catalyst sample was investigated by a range of structural probes and the activity monitored under a number of different conditions. Theoretical calculations and DRIFTS were used to describe a plausible reaction mechanism. Overall, I find that the conclusions in the manuscript are not sufficiently supported. There are many instances where details are missing for assignments or methodologies. These concerns are detailed below and should be addressed before consideration for publication.

One of the main claim of the manuscript is, from page 4, that 'To the best of our knowledge, due to the exothermic nature of methanol synthesis, it is yet to be achieved via photothermal CO₂ hydrogenation without any external thermal source.' in reference to a result of methanol production at a rate of 31.2 μmol g⁻¹ h⁻¹ with a selectivity of 36.7% using S2 without an external thermal source. However, this

result doesn't agree with the activity shown in Figure 2e where methanol production is negligible < 230 C. In addition, the maximal rate of methanol production doesn't exceed 25 $\mu\text{mol g}^{-1} \text{h}^{-1}$ even at 300 C. How was a higher methanol production rate obtained at, presumably, room temperature? The experimental details are lacking.

On page 4, it states 'The solid-state H-NMR spectrum of S2 confirms the presence of hydride species, which could potentially enhance the absorption and reduction of CO₂ and CO surface species (Figure S8b). 29 ' The assignment of the H peaks at 0.2 - 0.6 ppm to a hydride is surprising considering that typical metal hydrides have signal < 0 ppm. Reference 29 does not show H NMR results nor does it refer to a hydride species. More conclusive support for the assignment of the H NMR peak to a hydride species is needed.

It is mentioned a number of times that the thermodynamic equilibrium is surpassed/overcome. However, it is not mentioned what the expected results would be if the reaction was under thermodynamic equilibrium (and presumably only reactions 1 and 2 would be active?).

The results from the presented catalytic system were compared to simulations of the reactivity of CZA. There are a number of issues with how this was approached: 1) It isn't argued why CZA is an appropriate comparison; 2) The details of the simulations are not described. The only information given is that their were estimated by ASPEN. 3) Comparison with experimental data, not simulations, would highlight the special reactivity of S2 much better.

The term 'tandem' does not seem to be appropriate to describe the sequential RWGS and CO hydrogenation reactions. The applicable term is domino catalysis, as per DOI:10.1016/j.ccr.2004.05.012 . In a similar vein, there is nothing that indicates that this is an 'auto-accelerated' reaction pathway, as claimed in the conclusions.

On page 6, it states 'To further confirm the possibility of a photochemistry-facilitated mechanism, temperature-programmed photoluminescence was conducted over S2, and the prolonged lifetime of photo-excited electron-hole pairs was observed with increasing temperatures (Figure S14). ' Figure S14 shows a decrease in the photoluminescence intensity with increasing temperatures. It is unclear how the authors inferred longer lifetimes of photogenerated pairs from this data. In molecular fluorophores, lower photoluminescence intensity is observed at higher temperatures because the non-radiative processes get faster.

On page 6, it states 'Unlike a previous study, ...' but the study in question is not referenced.

Describing a band gap of 2.82 eV for S2 is unconvincing. The Tauc plot shows a curved trace with no obvious linear region. A bandgap maybe isn't even appropriate for this material.

In situ XAS experiments are described but no details are given regarding the light source other than it was an LED. More info are needed to give an idea of the irradiation conditions. This is from the SI where

it is stated 'A LED was placed on the side of the cell to illuminate the sample (maybe we need a picture or a drawing to illustrate the set up). ' The SI should be proofread and finalized. I agree that a picture or drawing would be beneficial.

Panels a and b of Figure 3 seem to be flipped. Panel a seems to be showing the results from 50% H₂ instead of 75% H₂.

Reviewer #3 (Remarks to the Author):

The authors presented an impressive work on methanol hydrogenation under ambient pressure using In₂O₃ with surface frustrated Lewis pairs. I support the acceptance of the work after the authors address several ambiguous discussions in their work.

1. In page 3, how are XRD, XPS and ICP-MS used to confirm the absence of metallic indium? In₃d spectrum was not cited in the main manuscript. In Figure S6, the In₃d peak showed a broadening at 443 eV which could be an indication of indium metal species.
2. As only a small volume of gas was analysed in the work, could the authors indicate the standard error of the collected data? In addition, are the batch and flow reaction conducted under similar pressure? Please indicate the operating pressure in Figure 2 for better clarification.
3. Please consider a more appropriate description to replace "methanol window".
4. What does the author mean by the simulation of CZA catalytic performance in page 7.
5. Why is the CH₃OH yield and selectivity higher under 50% H₂ compared to 75% H₂? How is this related to the tandem hydrogenation process proposed by the authors.
6. In page 8, what does the authors refer to by "such a phenomenon" in line 219? Please clarify.

REVIEWER COMMENTS

Reviewer #1 (Remarks to the Author):

The authors report the black indium oxide catalyst for photothermal CO₂ hydrogenation into CH₃OH at a low hydrogen concentration and ambient pressure. Although they employ both the experimental and theoretical methods to verify this way, the selectivity of CH₃OH is too low and the overall property is unattractive. Besides, they seemingly ignore the discussion about the effect of light relative to the heat for this process. Therefore, the quality of this work does not meet the high standard for “Nat. Commun.”. Moreover, there are several other issues that dampen my enthusiasm for this manuscript:

Author Reply: We regret that the referee might have misunderstood the novelty of the work presented in this manuscript. Particularly, the referee might have greatly underestimated the challenge to achieve effective methanol production from CO₂ hydrogenation with reasonably high rate and selectivity at atmospheric pressure.

Although Cu/ZnO based catalysts could achieve high methanol selectivity at high pressures (MPa), the methanol selectivity at low to atmospheric pressure is poor (usually <0.1%). To emphasize this point, the commercial Cu/ZnO/Al₂O₃ catalyst is prepared and tested in our system under the same conditions (CO₂:H₂ = 3:1, light, 30 psi). As shown in Figure. R1, CZA cannot produce any methanol under these reaction conditions. **These results clearly reveal the impressive methanol selectivity of our catalysts.**

Figure R1. (a) Photothermal catalytic performance of Cu/ZnO/Al₂O₃(CZA), S1 and S2 in the batch reactor, and (b) the corresponding methanol selectivity. Reaction conditions: H₂/CO₂ ratio = 3:1 with 30 Psi and a 300 W Xe lamp for a duration of 0.5 h without external heat.

Q1. Fig. S2 show the SEM images and size distributions of S1 and S2. There is a good deal of particles in the SEM images and how their size distributions are determined?

Author Reply: We measured particle size with ImageJ software, and the process as follows:

1. Import the targeted image to ImageJ software
2. Use “Line” tool to measure the length of scale bar in the image as precise as possible. Then

set the known distance and unit of length as the same as the scale bar.

3. Use “Line” tool to measure the major and minor lengths of one particle. Repeat the same procedure to get sizes as many as possible.

4. Analyze the measured length using statistical method to show the distribution.

Q2. Where are the original XPS spectra for S1 and S2 in Fig. 1C? Whether the XPS spectra in the manuscript match with their original XPS spectra?

Author Reply: We thank the reviewer for this suggestion. As shown in the picture, the XPS spectra of S1 and S2 match with their raw line spectra very well (The dashed line is original XPS spectra, and the solid line is our fit to the results).

Figure R2 O 1s spectra of S1 and S2 (The dashed line is original XPS spectra, and the solid line is our best fit results).

Q3. Can the positions of the valence-band maximum and conduction-band minimum meet the theoretical potential of CO₂/CH₃OH?

Author Reply: The gas-phase photothermal catalytic CO₂ hydrogenation towards methanol synthesis ($\text{CO}_2 + \text{H}_2 \rightarrow \text{CH}_3\text{OH} + 2\text{H}_2\text{O}$; $E=0.85 \text{ V}$) is different from the photocatalytic liquid-phase CO₂ reduction system ($\text{CO}_2 + 2\text{H}_2\text{O} \rightarrow \text{CH}_3\text{OH} + 2\text{O}_2$; $E = 1.21 \text{ V}$).

In the liquid system, as a multi-electron reduction process, CO₂ reduction usually requires the participation of protons, also known as proton coupled electron transfer (PCET) processes, in addition to a sacrificial electron agents (such as alcohol, TEOA).

Gas-phase photocatalytic CO₂ hydrogenation does not require the PCET process and sacrificial agents. In the case of the new black form of indium oxide described in this paper, the process has a significant thermal energy component rather than being a process driven by purely solar energy driven. Thus, the use of traditional energy band theory (valence band, conduction band, and bandgap) cannot explain this phenomenon adequately and is not the correct way to think our experimental phenomena. Part of the solar energy is involved the photocatalytic reaction and the remaining solar energy is transferred to thermal energy via the photothermal effect.

Furthermore, Ultraviolet photoelectron spectroscopy (UPS) has been conducted to define the photocatalysts electronic band-structure in **Figure R3**. The conduction band and valence band positions of -3.64 eV and -7.05 eV were obtained for S2, both of which meet the theoretical potential for the redox couples CO₂/CH₃OH and CO/CH₃OH.

Figure R3. Ultraviolet photoelectron spectroscopy measurement of E_{cutoff} (a) and E_f (b) establish conduction band and valence band positions of -3.55 eV and -7.04 eV for S1; and -3.64 eV and 7.05 eV for S2. (c) Energy band structure of samples.

Q4. What is the opposite half reaction relative to the CO₂ hydrogenation into CH₃OH?

Author Reply: If the CO₂ reduction occurred in the aqueous phase, which is not the case for our gas phase process, the half reaction would be:

However, in our gas-phase system, the reactions are $\text{CO}_2 + \text{H}_2 \rightarrow \text{CO} + \text{H}_2\text{O}$ and $\text{CO} + 2\text{H}_2 \rightarrow \text{CH}_3\text{OH}$, so it is not correct to assign a half-reaction as in the aqueous phase.

Q5. Fig. S11 shows the chromatography spectra signal of CH₃OH. Is it a gas form or liquid form?

Author Reply: The CH₃OH is in the gas phase that flows together with other products to the gas chromatograph (GC). The reactor is connected directly to the GC, and all results are obtained “online”. The CH₃OH and CO are the only detectable products.

Q6. The isotope-labeled mass spectrum in Fig. S15 is incomplete. Are there the peaks of other smaller species (¹³CH₃, ¹³CH₂, ¹³CH) and ¹³CO₂?

Author Reply: The peaks for other species are shown in **Figure R4**. The peaks of ¹³CH₃, ¹³CH₂, ¹³CO, and ¹³CO₂ proved unequivocally that methanol comes from CO₂ hydrogenation.

Figure R4. (a) Typical mass spectra with the m/z arranged 13-45. (b) The CH_3OH products when using the isotope-labeled $^{13}\text{CO}_2$ feedstock.

Q7. How the properties of photocatalytic CO_2 reduction by these two catalysts? The comparative experiments should be added.

Author Reply: We thank the reviewer for this excellent suggestion. Now, the self-prepared commercial samples $\text{Cu/ZnO/Al}_2\text{O}_3$ and the precursor In_2O_3 are used as the control samples. The photothermal catalytic performance is shown in **Figure R1**, whereas S1 shows no catalytic performance, both CZA and S2 could yield CO , and only the black indium oxide could produce methanol. This result implies the black indium oxide is an excellent methanol maker.

Q8. The authors should compare the property of CO_2 hydrogenation in this work with other reported references (10.1038/s41565-018-0089-z; 10.1126/sciadv.aaz2060).

Author Reply: We thank the reviewer for this suggestion.

However, a proper comparison is challenging, since **the innovation and significance of our work is to show the tandem methanol synthesis pathway over black indium oxide produces methanol at low H_2 concentration with high methanol selectivity.**

All previous work exhibits CO_2 hydrogenation towards methanol synthesis with 75% to 80% H_2 and our catalyst only needs 50% H_2 . Furthermore, it is also the first time to achieve methanol synthesis via a photothermal catalytic process in a batch system, which makes it difficult to compare with other work. Thus, we could only provide a rough comparison with all photocatalytic-related processes under different conditions, which is best not to add to the manuscript to avoid misunderstanding.

Table R1. Comparison of the catalytic performance with reported references.

Catalyst	Conditions	Methanol rate ($\mu\text{mol h}^{-1}\text{g}_{\text{cat}}^{-1}$)	Methanol selectivity	Reference
Photocatalysis				
$\text{rh-In}_2\text{O}_{3-x}(\text{OH})_y$	H_2 (6 ml/min) CO_2 (2 ml/min) 523 K 130 W Xe lamp	126	19.7%	1
$\text{LaNi}_{0.2}\text{Fe}_{0.8}\text{O}_3$	H_2O (16.7 mmol) CO_2 (6.0 mmol)	0.98	3.8%	2

	6 h 300 W Xe lamp ($\lambda > 420$ nm)			
CuZnGa-LDH/WO ₃	H ₂ O CO ₂ (3.5 Kpa) 323K 500 W Xe lamp	0.045	65%	3
Zn-Cu-Al LDH	H ₂ (1.67 mmol) CO ₂ (0.177 mmol) 500 W Xe lamp	0.2	25.64%	4
Zn-Cu-Ga LDH	H ₂ (1.67 mmol) CO ₂ (0.177 mmol) 500 W Xe lamp	0.17	68.27%	4
Zn-Ga-CO ₃ LDH	0.4 MPa H ₂ (0.28 MPa) CO ₂ (0.12 MPa) 500 W Xe lamp	0.19	48.71%	5
[Zn _{1.5} Cu _{1.5} Ga(OH) ₈] ⁺² [Cu(OH) ₄] ²⁻	H ₂ (1.67 mmol) CO ₂ (0.177 mmol) 423 K 500 W Xe lamp	0.49	87.50%	6
Ag/Zr ₃ Ga LDH	~0.8 MPa H ₂ (0.28 MPa) CO ₂ (0.12 MPa) 500 W Xe lamp	0.1	3.17%	7
Cu/TiO ₂	~0.8 MPa H ₂ (0.28 MPa) CO ₂ (0.12 MPa) 500 W Xe lamp	0.056	2.92%	7
In ₂ O _{3-x} (OH) _y nanocrystal superstructure	H ₂ (6 ml/min) CO ₂ (2 ml/min) 523 K 140 W Xe lamp	63.73	53.03%	8
AuCu/g-C ₃ N ₄	CO ₂ and H ₂ O (l) 120 °C 300 W Xe lamp ($\lambda > 420$ nm)	0.89	93.1%	9
0.3%Bi-In ₂ O _{3-x} (OH) _y	H ₂ (15 Psi) CO ₂ (15 Psi) Batch reactor 12 h 423 K 1000 W Xe lamp	0 (only CO about 1.32 $\mu\text{mol h}^{-1}\text{g}_{\text{cat}}^{-1}$)	0	10
H _z In ₂ O _{3-x} (OH) _y	H ₂ (3 ml/min) CO ₂ (1 ml/min) 523 K 300 W Xe lamp	7.54 (after 75 h)	49.21%	This work
H _z In ₂ O _{3-x} (OH) _y	H ₂ (22.5 Psi) CO ₂ (7.5 Psi) Batch reactor 0.5 h 300 W Xe lamp	31.23	36.8%	This work

Reviewer #2 (Remarks to the Author):

The manuscript by Zhang et al. describes the a modified In₂O₃ catalyst that is capable of synthesizing methanol from CO₂ (or CO) and H₂. The modified In₂O₃ is black and demonstrates photothermal activity under Xe lamp irradiation. The catalyst sample was investigated by a range of structural probes and the activity monitored under a number of different conditions. Theoretical calculations and DRIFTS were used to describe a plausible reaction mechanism. Overall, I find that the conclusions in the manuscript are not sufficiently supported. There are many instances where details are missing for assignments or methodologies. These concerns are detailed below and should be addressed before consideration for publication.

Author Reply: We thank the reviewer for the positive appraisal of the work reported in our paper and deeply appreciate the chance to respond to the comments voiced below.

Q1. One of the main claim of the manuscript is, from page 4, that 'To the best of our knowledge, due to the exothermic nature of methanol synthesis, it is yet to be achieved via photothermal CO₂ hydrogenation without any external thermal source.' in reference to a result of methanol production at a rate of of 31.2 $\mu\text{mol g}^{-1} \text{h}^{-1}$ with a selectivity of 36.7% using S2 without an external thermal source. However, this result doesn't agree with the activity shown in Figure 2e where methanol production is negligible < 230 C. In addition, the maximal rate of methanol production doesn't exceed 25 $\mu\text{mol g}^{-1} \text{h}^{-1}$ even at 300 C. How was a higher methanol production rate obtained at, presumably, room temperature? The experimental details are lacking.

Author Reply: We thank the reviewer for carefully checking of our manuscript.

As shown in figure captions, we showed two types of results in this manuscript that were obtained from the batch reactor and flow reactor. The batch reactor provided us the opportunity to initiate the photothermal catalysis without an external heat supply (Fig. 2a). The flow reactor usually requires external thermal energy and provides a platform to study the effect of thermal energy, light energy and obtain detailed information for reaction kinetics (Fig. 2c and 2e).

Furthermore, the batch reactor indicates the reactants in the reactor will not flow in and out, the contact time between reactants and catalyst is the same as the reaction time and usually result in very different catalytic performance to the flow reactor. In detail, the difference between results from batch and flow reactor indicate the advantages of photothermal catalytic strategy in which the black indium oxide could convert some of the incident solar energy into thermal energy and conduct the CO₂ hydrogenation without an external thermal source. Since the flow reactor has a lower reactant to catalyst contact time than the batch system, the reaction performance and kinetic could be different. Moreover, based on the results from the flow reactor, we can identify the interesting trend of activation energy, the possible mechanism of light and two distinct active sites.

To avoid misunderstanding of the results from different systems, more detailed experimental information is added into figure caption and **Supporting Information**.

The gas-phase photothermal catalytic evaluations were conducted in a custom-built 1.5 mL stainless steel batch reactor with a fused silica viewport sealed with Viton O-rings. The reactor was evacuated using an Alcatel dry pump prior to being purged with the reactant H₂ gas (99.9995%). After purging

the reactor, it was filled with a 3:1 stoichiometric mixture of H₂ (99.9995%) and CO₂ (99.999%) until the total pressure reached 30 psi. Reactors were irradiated with a 300 W Xe lamp for a duration of 0.5 h without external heat.

Gas-phase flow reactor measurements were carried out in a fixed-bed tubular reactor (3 mm outer diameter and 2 mm inner diameter). During the reaction, H₂ (Praxair 99.999%) and CO₂ (Praxair 99.999%) were flowed in a 1:1 ratio at a total volumetric flow rate of 4 ml/min. For photocatalytic rate measurements, the reactor was irradiated with a 300 W Newport Xe lamp. Product gases were analyzed using FID and TCD installed in an SRI-8610 gas chromatograph equipped with a 3 in. Mole Sieve 13a and a 6 in. Haysep D column.

Q2. On page 4, it states 'The solid-state H-NMR spectrum of S2 confirms the presence of hydride species, which could potentially enhance the absorption and reduction of CO₂ and CO surface species (Figure S8b). 29 ' The assignment of the H peaks at 0.2 - 0.6 ppm to a hydride is surprising considering that typical metal hydrides have signal < 0 ppm. Reference 29 does not show H NMR results nor does it refer to a hydride species. More conclusive support for the assignment of the H NMR peak to a hydride species is needed.

Author Reply: We appreciate reviewer's suggestion. Now the reference 26 is also added into this sentence. The reference 26 (*Angew. Chem. Int. Ed.* 2019, 58, 9501-9505.) show similar H-NMR results to this work with a possible hydride located in the positive chemical shift region. Based on Lenz's law, hydride is electron rich and is more shielded than reference adamantane, therefore typically hydrides occur at high field, which is low frequency. When hydride is bonded to a more electronegative element like indium (electronegativity: In>Ca), the hydride in In-H⁻ should be less shielded than Ca-H⁻ (≈5 ppm) and therefore can occur at a less positive chemical shift than Ca-H⁻, as we observe (In-H⁻, ≈0.6 ppm). To further support this proposal, another more electronegative metal, Ag was reported able to dissociate H₂ in an Ag-zeolite system and provide a new Ag-H peak at -0.1ppm (*J. Phys. Chem. C* 2013, 117, 7690-7702.), which is more negative than the one we observed. The reference 29 (*J. Am. Chem. Soc.* 2016, 138, 1206-1214.) demonstrates that the surface frustrated Lewis pairs (FLPs) created by a Lewis acidic coordinately unsaturated surface indium site proximal to an oxygen vacancy and hence enhanced activity of defected indium oxide surfaces for the gas-phase reverse water gas shift reaction, CO₂ + H₂ + hν → CO + H₂O in the light compared to the dark. In **Figure R4**, the hydride is formed by the heterolytic dissociation of H₂ on the surface FLPs in In₂O_{3-x}(OH)_y.

On the other hand, from the H₂ temperature-programmed desorption (H₂-TPD), the three distinct absorption peaks for S2 could correspond to the physical adsorption of H₂ on the sample surface at lower temperatures (<150 °C), desorption of surface hydrides (200-300 °C), and surface protonated

hydroxides (>300 °C) at higher temperatures (**Figure S9a**), respectively. The H₂-TPD results thereby confirm the presence of hydride species.

Figure R4 the heterolysis pathway reaction of the dissociation of H₂ on the surface FLPs in In₂O_{3-x}(OH)_y.

Q3. It is mentioned a number of times that the thermodynamic equilibrium is surpassed/overcome. However, it is not mentioned what the expected results would be if the reaction was under thermodynamic equilibrium (and presumably only reactions 1 and 2 would be active?).

Author Reply: We thank the reviewer's suggestion. First, it is known that the CO₂ hydrogenation for methanol synthesis is accompanied by the reverse water gas shift (RWGS) reaction. The methanol synthesis process is an exothermic process and RWGS is an endothermic process. As shown in **Figure R5a**, the standard Gibbs free energy for CO₂ hydrogenation towards methanol increases with temperature, and that of RWGS reaction decreases with temperature, which indicates RWGS favors high temperatures, and the production of methanol favors lower temperatures. As a result, with the increasing temperature, RWGS will dominate the overall process and result in low methanol selectivity. A well-known example is CZA, which cannot produce any methanol in photothermal catalytic batch system. Higher pressure and higher concentration of H₂ can also result in high methanol selectivity, **Figure R5b-c**.

The ASPEN chemical engineering software is used to simulate the thermodynamic equilibrium under our reaction condition (250 °C, atmospheric pressure, flow reactor), the CO selectivity is about 99.89% for the RWGS (reaction 1) thermodynamic equilibrium, and 0.11% methanol selectivity for reaction 2 thermodynamic equilibrium. For the batch system, the selectivity of CO and methanol are 99.01% and 0.99%, respectively. From the experimental perspective, the prepared CZA no methanol can be detected from the catalytic system. On the contrary, the S2 could produce methanol via photothermal catalysis in the batch system with a selectivity of ~32% and ~30% in the flow reactor. The high methanol selectivity indicates the thermodynamic equilibrium has been surmounted via the tandem methanol synthesis pathway.

Figure R5 the Gibbs free energy of RWGS and MeOH product. (b) The relationship between the methanol yield and pressure. (c) The CO/methanol selectivity with the different ratio of H₂/CO₂ under the thermodynamic equilibrium.

Q4. The results from the presented catalytic system were compared to simulations of the reactivity of CZA. There are a number of issues with how this was approached: 1) It isn't argued why CZA is an appropriate comparison; 2) The details of the simulations are not described. The only information given is that they were estimated by ASPEN. 3) Comparison with experimental data, not simulations, would highlight the special reactivity of S2 much better.

Author Reply: The Cu/ZnO/Al₂O₃ (CZA) is a well-known catalyst for CO₂ hydrogenation towards methanol synthesis. A direct comparison between CZA and our catalyst helped to evaluate the catalyst. The prepared black indium oxide with SFLPs resulted in excellent methanol making tendency, which enabled the methanol synthesis at low H₂ concentrations.

We agree that the result should be compared with experimental data, the CZA is prepared and tested in a photothermal batch reactor. The experimental data for CZA is added into **Figure R5**. S2 is the only catalyst that we have found to produce methanol via the photothermal batch system. In addition, details of the simulations have been added into **Supporting Information**.

Figure R6. (a) Photothermal catalytic performance of Cu/ZnO/Al₂O₃(CZA), S1 and S2 in the batch reactor, and (b) corresponding to their methanol selectivity. Reaction conditions: H₂/CO₂ ratio = 3:1 with 30 Psi and a 300 W Xe lamp for a duration of 0.5 h without external heat.

Q5. The term 'tandem' does not seem to be appropriate to describe the sequential RWGS and CO hydrogenation reactions. The applicable term is domino catalysis, as per DOI:10.1016/j.ccr.2004.05.012 . In a similar vein, there is nothing that indicates that this is an 'auto-accelerated' reaction pathway, as claimed in the conclusions.

Author Reply: We appreciate reviewer's valuable suggestions. It helps us a lot in the understanding of tandem and domino catalysis.

We read this reference carefully (*Coordination chemistry reviews* 248.21-24 (2004): 2365-2379.), which contains the definitions of tandem catalysis and domino catalysis. As shown in the reference Figure 1, the domino catalysis emphasizes the realization of multiple transformations through a **single catalytic mechanism**, while the tandem catalysis focuses on **multiple catalytic mechanisms**.

In our work, based on the catalytic performance, estimated activation energies, DRIFTS and DFT simulation, two types of active sites and two possible mechanisms were proposed, which involve surface FLPs and oxygen vacancy, the former corresponds to reverse water gas shift (RWGS) catalytic mechanism and the latter for the methanol synthesis (CO hydrogenation). Therefore, we believe tandem could be a more suitable word for description.

We agree that the term 'auto-accelerated' is not suitable and replace the sentence to “The tandem process shifts the conventional competing RWGS and methanol synthesis process into a combined reaction pathway.” The “combined reaction pathway” here indicates the combination of RWGS and CO hydrogenation for methanol synthesis.

Fig. 1. Flowchart for classification of one-pot processes involving sequential elaboration of an organic substrate via multiple catalytic transformations.

Q6. On page 6, it states 'To further confirm the possibility of a photochemistry-facilitated mechanism, temperature-programmed photoluminescence was conducted over S2, and the prolonged lifetime of photo-excited electron-hole pairs was observed with increasing temperatures (Figure S14). ' Figure S14 shows a decrease in the photoluminescence intensity with increasing temperatures. It is unclear how the authors inferred longer lifetimes of photogenerated pairs from this data. In molecular fluorophores, lower photoluminescence intensity is observed at higher temperatures because the non-radiative processes get faster.

Author Reply: In photoluminescence (PL) incident light excites the valence band electrons to a higher electronic excited state, and leave charge balancing holes in the valence band, some excited state electrons will also relax non-radiatively to a lower energy level, recombine with the holes, and emit light.

Photoluminescence is widely used in photocatalysis to characterize the recombination of photogenerated electrons and holes, and evaluate their lifetime. The higher the electron-hole recombination rate, the lower the separation rate. For steady-state PL spectra, it is generally believed that higher peak intensity indicates a higher recombination rate, and lower separation efficiency of photogenerated electrons and holes. (*ACS Nano* 2018, 12, 5551-5558. *Applied Catalysis B: Environmental* 2021, 291, 120104. *Advanced Functional Materials* 2021, 31(18), 2100816.)

The PL spectrum in semiconductor materials is well-documented to be a very effective technique to reveal the recombination processes of the photogenerated e^- and h^+ pairs, trapping, and diffusion. When a semiconductor is excited by bandgap energy light, the excited e^- will transfer from the VB to the CB and form photogenerated h^+ in the VB. In the meantime, the photoexcited e^- will return to the VB and combine with photogenerated h^+ . In this process, the excess energy could be released as direct band–band photoluminescence emission. Thus, the higher PL intensity represents higher recombination efficiency of photogenerated e^- and h^+ . (*Chemical Engineering Journal* 2021, 422,

Because non-radiative processes get faster with temperature, lower photoluminescence intensity could be observed at higher temperatures. This process will happen to both S1 and S2, and lower their PL intensity simultaneously and result in overlapping PL spectra. Thus, we take S1 as the reference, and confirmed that S2 exhibit lower PL intensity, which represents lower recombination efficiency and prolonged lifetime of electron-hole pairs (**Figure R7**).

In response to the referee's concern, we have added the following sentence to manuscript:

“To further confirm the possibility of a photochemistry-facilitated mechanism, temperature-programmed photoluminescence was conducted, and the lower recombination efficiency and prolonged lifetime of photo-excited electron-hole pairs over S2 was observed.¹¹⁻¹³ (Figure S14)”.

Figure R7. Photoluminescence spectra of S1 and S2 for different temperature ranges (200–290 °C).

Q7. On page 6, it states 'Unlike a previous study, ...' but the study in question is not referenced.

Author Reply: We thank the referee for these valuable suggestions, which helped us improve the quality of the manuscript. The related reference is added into the manuscript.

“Unlike the previous studies, the methanol rate is notably inhibited at high temperatures^{1-2, 30}”.

1 Jiang, X., Nie, X. W., Guo, X. W., Song, C. S. & Chen, J. G. G. Recent Advances in Carbon Dioxide Hydrogenation to Methanol via Heterogeneous Catalysis. *Chem. Rev.* **120**, 7984-8034 (2020).

2 Zhong, J. W. et al. State of the art and perspectives in heterogeneous catalysis of CO₂ hydrogenation to methanol. *Chem. Soc. Rev.* **49**, 1385-1413 (2020).

30 Wang, L. et al. Photocatalytic Hydrogenation of Carbon Dioxide with High Selectivity to Methanol at Atmospheric Pressure. *Joule* **2**, 1369-1381 (2018)

Q8. Describing a band gap of 2.82 eV for S2 is unconvincing. The Tauc plot shows a curved trace with no obvious linear region. A bandgap maybe isn't even appropriate for this material.

Author Reply: We recalculated the band gap of the samples with the Tauc plot method in **Figure R8**, 3.49 eV for S1 and 3.39 eV for S2. The ultraviolet photoelectron spectroscopy (UPS) is conducted to measure the band-structure and give conduction band and valence band positions respectively -5.08 eV and -7.05 eV for S2 and the bandgap can be estimated as 3.39 eV, agrees well with the estimation of Tauc plot (**Figure R3**).

Figure R8. (a) The UV-VIS-NIR spectra of S1 and S2. (b) The method of bandgap energy determination from the Tauc plot.

Q9. In situ XAS experiments are described but no details are given regarding the light source other than it was an LED. More info are needed to give an idea of the irradiation conditions. This is from the SI where it is stated 'A LED was placed on the side of the cell to illuminate the sample (maybe we need a picture or a drawing to illustrate the set up). ' The SI should be proofread and finalized. I agree that a picture or drawing would be beneficial.

Author Reply: We thank the reviewer for this valuable suggestion. The detailed information of in situ XAS is added in Supporting Information, and the picture of the in situ XAS cell is shown in **Figure R8** and also added into the manuscript.

Figure R9. The picture of in situ XAS cell.

Q9. Panels a and b of Figure 3 seem to be flipped. Panel a seems to be showing the results from 50% H₂ instead of 75% H₂.

Author Reply: We thank the reviewer for carefully checking, and the oversight has been corrected.

Figure 3. Stability test and *in-situ* DRIFTS analysis. 75-hours continuous stability test for S2 at 250 °C with light under (a) 75% H₂ and (b) 50% H₂. Reaction conditions: atmospheric pressure, H₂/CO₂ ratio of 1:1 or 3:1, a flow rate of 4 mL min⁻¹ and light intensity of ~6 suns. (c) *In-situ* DRIFTS of S2 with CO₂ and H₂ (1:1) and (d) with CO and H₂ (1:1). All DRIFTS spectra are subtracted by the background signal of S2 obtained under He. α : formate; β : methoxy; γ : methanol; δ : H₂O.

Reviewer #3 (Remarks to the Author):

The authors presented an impressive work on methanol hydrogenation under ambient pressure using In_2O_3 with surface frustrated Lewis pairs. I support the acceptance of the work after the authors address several ambiguous discussions in their work.

Author Reply: We thank the reviewer for the positive appraisal of the work reported in our paper and deeply appreciate the chance to respond to the comments voiced below.

Q1. In page 3, how are XRD, XPS and ICP-MS used to confirm the absence of metallic indium? $\text{In}3d$ spectrum was not cited in the main manuscript. In Figure S6, the $\text{In}3d$ peak showed a broadening at 443 eV which could be an indication of indium metal species.

Author Reply: The XRD can identify the crystalline structure, such as crystalline In_2O_3 and metallic In. The major signal of metallic In in the PXRD appears at 32.8° , missing in the PXRD patterns of S2 (Figure R10).

The XPS results show that the two peaks at 451.6 eV and 444.0 eV correspond to $\text{In}3d_{5/2}$ and $\text{In}3d_{3/2}$, respectively, and both belong to In_2O_3 . In Figure S6, the $\text{In}3d$ peak showed broadening which could be caused by the surface hydride species (including FLPs) that increase the electron density at surface indium species. Furthermore, the broadened $\text{In}3d$ peak is known and seen in our previous studies (Nat. Commun. 2020, 11, 2432, Fig. S2) where metallic In is not present.

In addition, the *in-situ* X-ray absorption near edge structure (XANES) spectrum for S2 was recorded in various atmospheres and different temperatures and also confirmed the absence of metallic indium (Figure 2d).

Figure R10. XRD pattern of S2 and mixed In- In_2O_3 samples.

Figure 2d. In-situ XANES of S2 catalyst under different reaction conditions.

Q2. As only a small volume of gas was analyzed in the work, could the authors indicate the standard error of the collected data? In addition, are the batch and flow reaction conducted under similar pressure? Please indicate the operating pressure in Figure 2 for better clarification.

Author Reply: We thank the referee for the valuable suggestions, which helped us improve the quality of the manuscript. The estimated standard deviation of the data and the pressure information are added.

Photothermal catalytic performance of S1 and S2 in the batch reactor. Reaction conditions: H_2/CO_2 ratio = 3:1 with 30 Psi and light intensity of 20 suns for a duration of 0.5 h without external heat. Conditions for flow measurement: atmospheric pressure, H_2/CO_2 ratio = 1:1 or 3:1 with a total flow rate of 4 mL min^{-1} and light intensity of 6 suns.

Figure 2. Catalytic performance of the samples. (a) Photothermal catalytic performance of S1 and S2 in the batch reactor. Reaction conditions: H₂/CO₂ ratio = 3:1 with 30 Psi and light intensity of 20 suns for a duration of 0.5 h without external heat. (b) Methanol selectivity with different H₂: CO₂ ratios; D indicates dark, L indicates light, and B indicates batch reactor with 30 Psi. The methanol equilibrium selectivities were estimated by ASPEN. (c) CO rate of S2 in a flow reactor under light/dark conditions and (d) the corresponding Arrhenius plot, (e) methanol rate of S2 and (f) the corresponding Arrhenius plot. Conditions for flow measurement: atmospheric pressure, H₂/CO₂ ratio = 1:1 with a total flow rate of 4 mL min⁻¹ and light intensity of 6 suns.

Q3. Please consider a more appropriate description to replace “methanol window”.

Author Reply: We thank the reviewer for raising this important question. The methanol window is removed.

Q4. What does the author mean by the simulation of CZA catalytic performance in page 7.

Author Reply: We thank the reviewer for raising this important question. The Cu/ZnO/Al₂O₃ (CZA) is an efficient catalyst for CO₂ hydrogenation towards methanol and is also considered to be the standard industry catalyst for large-scale CO₂ hydrogenation. Therefore, a direct comparison between CZA and our catalyst could help us to evaluate and compare their performance. As suggested by the other reviewers, the experimental catalytic performance of CZA is also added into the manuscript and compared with our catalyst, **Figure 2a**. As a result, only S2 is able to yield

methanol in a photothermal batch system which confirmed S2's impressive capacity for making methanol. In addition, the methods used for the simulations has been added into **Supporting Information**.

Q5. Why is the CH₃OH yield and selectivity higher under 50% H₂ compared to 75% H₂? How is this related to the tandem hydrogenation process proposed by the authors.

Author Reply:

We would like to apologize for the oversight mislabelling of (a) and (b) in previous **Figure 3**. The updated Fig. 3 and its caption are shown below.

Figure 3. Stability test and *in-situ* DRIFTS analysis. 75-hours continuous stability test for S2 at 250 °C with light under (a) 75% H₂ and (b) 50% H₂. Reaction conditions: atmospheric pressure, H₂/CO₂ ratio of 1:1, a flow rate of 4 mL min⁻¹, and light intensity of ~6 suns. (c) *In-situ* DRIFTS of S2 with CO₂ and H₂ (1:1) and (b) with CO and H₂ (1:1). All DRIFTS spectra are subtracted by the background signal of S2 obtained under He. α: formate; β: methoxy; γ: methanol; δ: H₂O.

Based on the updated Fig. 3, the S2 exhibits a higher methanol rate and selectivity with 75% H₂ than the one with 50% H₂. Even after 70 hours, S2 with 75% H₂ exhibits higher methanol selectivity and a similar methanol rate than that of 50% H₂. The different **inhibition** processes and similar stabilized methanol rates could be caused by the different reaction mechanisms.

In Fig. 3a-b, for 75% H₂ concentration, both CO₂ + H₂ → H₂O + CO and CO₂ + 3H₂ → CH₃OH + H₂O yield water and result in the deactivation of CO and methanol production. For 50% H₂, only CO₂ + H₂ → H₂O + CO yields water and deactivates the RWGS process, whereas methanol remains stable and implies a water-free methanol production process. In this case, the CO hydrogenation (CO + 2H₂ → CH₃OH) could be the potential pathway for water-free methanol production.

To confirm this idea, the kinetic study, DRIFTS (both CO₂/H₂ and CO/H₂ systems) and DFT simulation were conducted and resulted in a reasonable tandem pathway including the RWGS reaction (CO₂ + H₂ → H₂O + CO) and CO hydrogenation (CO + 2H₂ → CH₃OH).

To amplify, the two-stage convex Arrhenius plots indicate different kinetic controlled pathways,^{14,15} whereas at lower temperatures, the reaction mainly depends on the activation and the dissociation of CO₂ molecules, and at higher temperatures, it mainly depends on the diffusion of reaction intermediates, such as *CO.^{14,16,17} In Fig. 4c-d, a 50% H₂ and CO *in-situ* DRIFTS measurement was conducted. Despite the presence of methanol-related peaks at 2800 cm⁻¹ – 2900 cm⁻¹^{18,19} and methoxy species at 3000 cm⁻¹ – 3100 cm⁻¹, the absence of typical formate species at 3200 cm⁻¹ and 1580 cm⁻¹ imply that the formate species could be the intermediate species for RWGS rather than methanol synthesis. It is worth noting that the peak at 1520 cm⁻¹ could correspond to the HOCO* species.²⁰ Since the DRIFTS was conducted in CO and H₂ atmosphere, the HOCO* was not supposed to be formed unless the In-OH group assisted SFLPs addition reaction was conducted. Moreover, the CO temperature-programmed desorption (CO-TPD) shown in **Figure S21** indicates strong CO adsorption on S2 surface. This strong CO adsorption could further benefit the subsequent CO hydrogenation in tandem methanol synthesis. The DFT simulation shows that upon CO₂ adsorption on the protonated and hydridic SFLPs sites, the endothermic RWGS reaction occurred on the surface. Both carbonate and formate were possible intermediates, and the *in-situ* DRIFTS experimentally identified the latter. Desorption of In-bonded CO to regenerate catalytic SFLPs sites should overcome an energy barrier of 0.84 eV, which is consistent with the high-temperature desorption peaks in the CO-TPD spectra, where some of the CO is difficult to desorb from the S2 surface and further be involved in the tandem reaction (methanol synthesis).

Q6. In page 8, what does the authors refer to by “such a phenomenon” in line 219? Please clarify.

Author Reply: We have replaced “Such a phenomenon” to “**The different deactivation processes**”, indicating the different deactivating pathways to CO and methanol with 50% and 75% H₂ (detail in Q5), whereas both CO and methanol were deactivated under 75% H₂ and only CO was deactivated under 50% H₂.

References

- 1 Yan, T. *et al.* Polymorph selection towards photocatalytic gaseous CO₂ hydrogenation. *Nat. commun.* **10**, 2521 (2019).
- 2 Zheng, D. *et al.* LaNi_xFe_{1-x}O₃ (0 ≤ x ≤ 1) as photothermal catalysts for hydrocarbon fuels production from CO₂ and H₂O. *J. Photochem. Photobiol. A-Chem.* **377**, 182-189, (2019).
- 3 Morikawa, M. *et al.* Photocatalytic conversion of carbon dioxide into methanol in reverse fuel cells with tungsten oxide and layered double hydroxide photocatalysts for solar fuel generation. *Catal Sci Technol* **4**, 1644-1651, (2014).
- 4 Ahmed, N., Shibata, Y., Taniguchi, T. & Izumi, Y. Photocatalytic conversion of carbon dioxide into methanol using zinc–copper–M(III) (M=aluminum, gallium) layered double hydroxides. *J. Catal.* **279**, 123-135, (2011).
- 5 Zhao, Y. *et al.* Layered Double Hydroxide Nanostructured Photocatalysts for Renewable Energy Production. *Adv. Energy Mater.* **6**, 1501974 (2016).
- 6 Ahmed, N., Morikawa, M. & Izumi, Y. Photocatalytic conversion of carbon dioxide into methanol using optimized layered double hydroxide catalysts. *Catal. Today* **185**, 263-269, (2012).
- 7 Navalón, S., Dhakshinamoorthy, A., Álvaro, M. & Garcia, H. Photocatalytic CO₂ Reduction using Non-Titanium Metal Oxides and Sulfides. *ChemSusChem* **6**, 562-577, (2013).
- 8 Wang, L. *et al.* Photocatalytic Hydrogenation of Carbon Dioxide with High Selectivity to Methanol at Atmospheric Pressure. *Joule* **2**, 1369-1381, (2018).
- 9 Li, P. Y. *et al.* Ultrathin porous g-C₃N₄ nanosheets modified with AuCu alloy nanoparticles and C-C coupling photothermal catalytic reduction of CO₂ to ethanol. *Appl. Catal. B-Environ.* **266**, 118618, (2020).
- 10 Dong, Y. *et al.* Tailoring Surface Frustrated Lewis Pairs of In₂O_{3-x}(OH)_y for Gas-Phase Heterogeneous Photocatalytic Reduction of CO₂ by Isomorphous Substitution of In³⁺ with Bi³⁺. *Adv. Sci.* **5**, 1700732, (2018).
- 11 Cui, L. *et al.* Constructing Highly Uniform Onion-Ring-like Graphitic Carbon Nitride for Efficient Visible-Light-Driven Photocatalytic Hydrogen Evolution. *ACS Nano* **12**, 5551-5558, (2018).
- 12 Li, B. *et al.* High-Throughput One-Photon Excitation Pathway in 0D/3D Heterojunctions for Visible-Light Driven Hydrogen Evolution. *Adv. Funct. Mater.* **31**, 2100816, (2021).
- 13 Shen, R. *et al.* In-situ construction of metallic Ni₃C@Ni core–shell cocatalysts over g-C₃N₄ nanosheets for shell-thickness-dependent photocatalytic H₂ production. *Appl. Catal. B-Environ.* **291**, 120104, (2021).
- 14 Chang, F. *et al.* Alkali and Alkaline Earth Hydrides-Driven N₂ Activation and Transformation over Mn Nitride Catalyst. *J. Am. Chem. Soc.* **140**, 14799-14806, (2018).
- 15 Truhlar, D. G. & Kohen, A. Convex Arrhenius plots and their interpretation. *P. Natl. Acad. Sci. USA* **98**, 848-851, (2001).
- 16 Fu, J. W., Jiang, K. X., Qiu, X. Q., Yu, J. G. & Liu, M. Product selectivity of photocatalytic CO₂ reduction reactions. *Mater. Today* **32**, 222-243, (2020).
- 17 Wang, J. Y. *et al.* CO₂ Hydrogenation to Methanol over In₂O₃-Based Catalysts: From Mechanism to Catalyst Development. *Acs Catal.* **11**, 1406-1423, (2021).

- 18 Wang, W. W., Qu, Z. P., Song, L. X. & Fu, Q. Probing into the multifunctional role of copper species and reaction pathway on copper-cerium-zirconium catalysts for CO₂ hydrogenation to methanol using high pressure in situ DRIFTS. *J. Catal.* **382**, 129-140, (2020).
- 19 Kunkes, E. L., Studt, F., Abild-Pedersen, F., Schlogl, R. & Behrens, M. Hydrogenation of CO₂ to methanol and CO on Cu/ZnO/Al₂O₃: Is there a common intermediate or not? *J. Catal.* **328**, 43-48, (2015).
- 20 Qi, Y. H. *et al.* Photoinduced Defect Engineering: Enhanced Photothermal Catalytic Performance of 2D Black In₂O_{3-x} Nanosheets with Bifunctional Oxygen Vacancies. *Adv. Mater.* **32**, 3915 (2020).

REVIEWER COMMENTS

Reviewer #1 (Remarks to the Author):

New Black Indium Oxide – Tandem Photothermal CO₂-H₂ Methanol Selective Catalyst

The authors have answered many of the questions. However, black In₂O₃ has been extensively studied in photothermal catalytic reactions and the novelty of this work should be further explained.

Additionally, there are still quite a few scientific problems and careless mistakes of this manuscript.

Therefore, this manuscript is ill-suited for the publication on Nature Communications. Some of the problems are listed below:

1. Measuring the size distribution in just one SEM image may result in significant errors, because one SEM image only display a very local morphology of the catalyst. The real particle size distribution may be very different.
2. In Fig.2a, what are the surface temperatures of S1 and S2 under illumination in the batch reactor? The difference in catalyst temperature could greatly affect their performances.
3. In Fig.3a and b, the performances of S2 decayed quite obviously during 75 h of test time. The authors should explain in detail the reasons for the performance decay. Additionally, the XPS measurements of used S2 samples should be conducted.
4. In Fig.S14, it is really hard to distinguish the PL lines of sample S1.
5. Lattice defects can be quite important for catalytic performances. However, little discussion was made with regard to the oxygen vacancies in the material. The authors should explain the effects of oxygen vacancies more carefully.
6. In Fig.4a, the authors declared that “The differential charge density map also evidenced electron density accumulation for the In-H bond and depletion from the hydroxyl group.”. However, it seems that the electron density is almost unchanged for the hydroxyl group.
7. In the DFT section, the authors declared that “Notably, this energy demand could be fully compensated by the heat release from previous H₂ adsorption and activation steps (E = -1.93 and -0.87 eV).”. Such a statement is really unscientific. No research has indicated that the energy released by previous reaction steps can help to overcome the energy barriers of subsequent reaction steps.
8. There are even no page numbers for the manuscript and supporting information.
9. At line 246, the authors stated that “The in-situ DRIFTS was used to track the intermediate species to understand the mechanism of methanol formation on S2 (Figure 4a-b, Figure S21).” Actually, the in-situ

DRIFTS spectra were in Fig.3 c, d and Fig.4 a, b were DFT results.

10. For the discussion of the in-situ DRIFTS studies, the authors should attribute different peaks to specific chemical groups rather than molecules.

Reviewer #2 (Remarks to the Author):

Many of my original concerns have been addressed, but I still have a few that need to be addressed.

Q1: In their response, the authors state 'Since the flow reactor has a lower reactant to catalyst contact time than the batch system, the reaction performance and kinetic could be different.' and 'Moreover, based on the results from the flow reactor, we can identify the interesting trend of activation energy, the possible mechanism of light and two distinct active sites.' However, the mechanism insight from flow reactor conditions is being presented as general. Based on the significant difference in reaction conditions and activity, it is possible that the reaction mechanism operating under batch and flow conditions are distinct. This should be clarified.

Q6: The interpretation of the PL data still seems idealized to me and lacks rigor. The sticking point is that the conclusion from 'Thus, we take S1 as the reference, and confirmed that S2 exhibit lower PL intensity, which represents lower recombination efficiency and prolonged lifetime of electron-hole pairs (Figure R7).' is only correct under the assumption that all recombination is *radiative*. I assume that the PL quantum yield of these materials aren't known, but I would expect it to be far lower than 100%. This means that it is safe to assume that there is some level of non-radiative recombination, typical when trapped states are involved in the recombination process. The fact that S2 has lower PL intensity (and a small decrease that appears to be < 5%) at all temperatures does not necessarily mean that the rate of recombination is lower. This data only allows to comment on the rate of *radiative* recombination. This means that a meaningful comment on the lifetime of electron-holes pair cannot be made. For example, the PL intensity of S2 could be lower than S1 because of an increased rate of non-radiative recombination and a shorter electron-hole lifetime.

Q8: I appreciate the additional UPS experiments to determine the band edges, but I still have concerns about a Tauc plot analysis to extract a band gap. I think it is telling that a difference in band gap of 0.5 eV can be obtained from the Tauc method (2.82 eV was revised to 3.39 eV for S2). To my eye a very wide range of values could be obtained as there is no clear, distinct linear section - the value will be very dependent on the specific range objectively chosen. A key fact to keep in mind is that, as prominently stated in the title, the material is black. A simple band structure with a roughly 3.4 eV band gap is incompatible with a black material. It's clearly absorbing light throughout the visible range, as well indicated in Figure R8/S4. The UPS does seem to indicate a separated VB and CB. The reflectance indicates a wide range of tail/trap states in order to have optical transitions throughout the UV-VIS-NIR spectral ranges.

Q9: There is still the words '(maybe we need a picture or a drawing to illustrate the setup)' in the SI, which is clearly a comment held over from previous edits. There is still no details given about the LED used to excite the samples for the in situ XAS. The wavelength range (or spectral output) of the LED and the irradiation power at the sample should be described.

Reviewer #3 (Remarks to the Author):

The revision is well done in most aspects. However, after the clarifications from the authors and reading through the manuscript, I do not find the methanol selectivity simulation component to be appropriate for inclusion in the final manuscript for the following reasons. I would suggest to remove this component and corresponding derived conclusions.

1. The simulation did not seem to account for the exothermic nature of the methanol synthesis reaction. In the revision, the author highlights the high selectivity towards CO but did not mention whether the simulation was performed in a non-adiabatic system. This may severely influence the calculated results in the case of CO and methanol selectivity.

2. Lack of simulation details which lead to the first suggested issue. It is also unclear to the reader how can the simulation be replicated for comparison purposes based on the provided details.

The simulation for calculating the methanol equilibrium yields and selectivities used the Gibbs reactor block in Aspen Plus V11 with the ideal NRTL property package. The specified reactions were RWGS and methanol synthesis. The feed used was 0.5:0.5; 0.67:0.33; 98 0.75:0.25 (H₂:CO) kmol/hr at 250 °C at different inlet pressure.

3. It was also unclear how could the simulation was assigned to CZA. It could be understood that the simulation was performed to obtain the thermodynamic equilibrium, but the obtained values does not correlate to the type of catalyst used. It is also misleading to state that a catalyst can be used to overcome thermodynamic equilibriums.

Reviewer #1:

The authors have answered many of the questions. However, black In_2O_3 has been extensively studied in photothermal catalytic reactions and the novelty of this work should be further explained. Additionally, there are still quite a few scientific problems and careless mistakes of this manuscript. Therefore, this manuscript is ill-suited for the publication on Nature Communications. Some of the problems are listed below:

Author Reply: We regret that the referee might once again have misunderstood the novelty of the work presented in this manuscript. According to the search results of Google Scholar and Web of Science, we could only find 3 papers related to black In_2O_3 for photothermal catalytic CO_2 hydrogenation, two of which come from Prof. Jinhua Ye's group and one from our group, all of which are near-unity for CO through reverse water gas shift reaction (RWGS).

To the best of our knowledge, there is no such report that could reasonably explain the high methanol selectivity of the black In_2O_3 at atmospheric pressure. In this work, two potential active sites (FLPs and V_o) were found that can promote the tandem reaction to synthesize methanol.

With the aforementioned points, we believe that our work has the novelty and quality to be accepted and published in Nature Communications.

1. Measuring the size distribution in just one SEM image may result in significant errors, because one SEM image only display a very local morphology of the catalyst. The real particle size distribution may be very different.

Author Reply: We agree with the reviewer's point. Unfortunately, we do not have a better technique to identify the particle size. Therefore, to minimize the experimental error and setup error, additional images (**Fig. R1**) of S1 and S2 were measured, and the mean particle size of S2 was found to be slightly larger than that of S1. XRD results can also confirm this (Debye-Scherrer equation: $D = K\lambda / (\beta \cos\theta)$; $D_{\text{S1}} = 67 \text{ nm}$, $D_{\text{S2}} = 78 \text{ nm}$).

Figure R1. (a)–(c) TEM images of S1 and the corresponding to mean sizes; (d)–(f) TEM images of S2 and the corresponding to mean sizes.

2. In Fig.2a, what are the surface temperatures of S1 and S2 under illumination in the batch reactor? The difference in catalyst temperature could greatly affect their performances.

Author Reply: This is an excellent question and all research groups in this field are trying to address it. We completely agree with the point that different temperatures can induce different performances. This is the actual theory for most of the photothermal catalysts.

To the best of our knowledge, the *in-situ* Raman spectroscopy could be the proper technique to tackle this challenging task. Unfortunately, the measurement of Raman spectra requires a strong laser, which dramatically increases the surface temperature of the black indium oxide. As a result, we could only observe the significantly increased temperature rather than the actual surface temperature. (<https://www.nature.com/articles/s41467-020-16336-z>, Fig. S8)

Based on the failure of the previous Raman experiment and special properties of this black indium oxide (same activation energy for RWGS with and without light), we tried to use the Arrhenius plot to estimate the real surface temperature from the kinetic aspect. As shown below, we have estimated that black In_2O_3 has a 16 °C photothermal advantage over the bulk temperature/environmental temperature in the flow reactor. (i.e. at 250°C, the surface temperature could be ~266°C)

The photothermal advantage for CO

In **Figure 2d**, the E_a also represents the catalytic mechanism for the whole reaction. Because E_a for dark and light are very close to each other, we assume that the black indium oxide only has the same reaction mechanism for RWGS with and without light, which means it is a photothermal/thermal process. Then the obtained CO rate with light can be used to calculate the temperature via the Arrhenius plot (dark).

$$y = -17.004x + 34.957 \text{ (dark)}$$

when $y = 3.433$ (ln(CO rate) at 250 °C with light)

$$x = 1.8539 \Rightarrow x = 1000/T \Rightarrow T = 539.4 \text{ K} \Rightarrow T = 266.4 \text{ }^\circ\text{C}$$

As a result, at 250 °C, the black indium oxide has a local temperature about 266 °C, given a 16 °C photothermal advantage (CO rate of 19.7 $\mu\text{mol g}^{-1} \text{ h}^{-1}$).

The photothermal advantage for methanol

In **Figure 2f**, the different temperature ranges with and without light also imply an about 20 °C photothermal advantage, which agrees well with the previous estimation of local temperature (about 16 °C). A similar method was applied to MeOH's Arrhenius plot for low temperature.

$$y = -20.218x + 39.418 \text{ (Dark-low)}$$

when $y = 1.9727$ (ln(MeOH rate) at 250 °C with light)

$$x = 1.8521 \Rightarrow x = 1000/T \Rightarrow T = 539.93 \text{ K} \Rightarrow T = 266.92 \text{ }^\circ\text{C}$$

The estimated local temperature agrees well with the previous estimation (about 16 °C photothermal advantage, MeOH rate of 5.14 $\mu\text{mol g}^{-1} \text{ h}^{-1}$). Therefore, the actual photothermal contribution for black indium oxide is about 16 °C under light conditions (lower than 250 °C).

3. In Fig.3a and b, the performances of S2 decayed quite obviously during 75 h of test time. The authors should explain in detail the reasons for the performance decay. Additionally, the XPS

measurements of used S2 samples should be conducted.

Author Reply: We thank the referee for their valuable suggestions which have helped us improve the quality of the manuscript.

In **Figure 3a**, the catalytic performance of S2 was measured with 75% H₂ (H₂:CO₂ = 3:1) at 250 °C, resulting in a high initial methanol rate of 14.92 μmol g⁻¹ h⁻¹ and fast decay that could be caused by the formation of water (reactions include: CO₂ + H₂ = CO + H₂O and CO₂ + 3H₂ = CH₃OH + H₂O).

When the concentration of H₂ was dropped to 50% H₂ at 250 °C, a very different catalytic performance (**Figure 3b**) was observed. Interestingly, as time passed, the CO rate was inhibited significantly and stabilized at 13.01 μmol g⁻¹ h⁻¹ (42% drop), while the methanol rate still remained 90% (6.48 μmol g⁻¹ h⁻¹).

It is well-known that the formation of H₂O will inhibit the methanol catalyst. The different decay processes indicate two distinct catalytic centers for the RWGS reaction and methanol synthesis. The water produced by the RWGS reaction could inhibit the CO rate. The negligible inhibition of methanol synthesis could be caused by the subsequent CO hydrogenation which did not produce additional water molecules (CO + 2H₂ = CH₃OH). Thus, with the help of *in-situ* DRIFTS and DFT simulation, a tandem reaction pathway on black In₂O₃ has been proposed.

The XPS results of the used S2 sample are shown in **Figure S7** and **Table S1**.

Figure S7. (a) In 2p spectra; (b) the O 1s spectra of used S2.

Table S1 Quantitative analysis for surface O from the O1s XPS.

	S1	S2	S2-used
In-O	75.77%	43.88%	47.46%
[O]	24.23%	44.87%	43.02%
In-OH	0	11.25%	9.52%

4. In Fig.S15, it is really hard to distinguish the PL lines of sample S1.

Author Reply: We appreciate reviewer's suggestion. Photoluminescence spectra of S1 and S2 have been redrawn, and the PL lines of samples S1 and S2 can be clearly distinguished.

Figure S15. (a) Summary photoluminescence spectra of S1 and S2 at different temperatures (200–290 °C). Separate PL spectra of (b) S1 and (c) S2.

5. Lattice defects can be quite important for catalytic performances. However, little discussion was made with regard to the oxygen vacancies in the material. The authors should explain the effects of oxygen vacancies more carefully.

Author Reply: The effects of oxygen vacancies ([O]) are explained several times in this work. We have marked the [O] related information as green.

- The oxygen vacancy is one of the active sites, capturing part of the produced CO (from RWGS reaction) and undergoing a tandem reaction to produce methanol.
- The SFLP sites (InOH•••In, **Figure S23d**) contain the [O] and an end-on hydroxyl group (OH) on [O]-vicinal indium atom, which is prone to heterolysis of H₂.
- DFT simulation results show that an exothermic conformation transition was observed when the CO (HO-In-[O]-In-CO) was trapped by [O] *via* an end-on mode (E = -0.17 eV in **Figure S24**). The nucleophilic oxygen atom refilled the [O] and the electrophilic carbon atom bonded to the OH, forming a bridging two-coordinated *sp* carbon (OCOH) species that was prone to be hydrogenated to *sp*³ hybridization. By comparison, higher energy barriers (E = 2.99 eV) were observed for methanol synthesis *via* “CO₂-H_xCO₂-CH₃OH” (x = 1, 2 and 3) on a pure [O]-laden In₂O_{3-x} surface (**Figure S25**).

Also, we provided further discussion in this revision as follow: It is worth noting that direct CO₂ hydrogenation to methanol on the [O] of In₂O₃ is also feasible, which has been studied by Ge et al.⁴⁴ and Sun et al.⁴⁵ recently. The pathway is believed to be “CO₂-H_xCO₂-CH₃OH” (x = 1, 2, and 3) with corresponding activation energy barrier (0.64–2.52 eV) associated with the polymorph, exposed facet of In₂O₃ and the position of the surface oxygen vacancy. The major difference between our SFLPs model and the [O] model is the involvement of an [O]-vicinal hydroxyl group. To understand this Lewis base OH-induced difference during CO₂-to-methanol conversion, we also calculated the free energy diagram on the OH-free [O] sites of the In₂O₃ (110) facet, which was constructed on the top of the SFLP model by abstracting the OH group. Distinct to the “downhill” methanol assembly in the SFLP-related pathway, energy barriers of 2.99 eV *via* an In-associated stepwise hydrogenation or 1.53 eV *via* a combined atomic and molecular hydrogenation pathway were observed (**Figure S25**). This result agreed with previous computational work, indicating our SFLP-associated tandem

RWGS-methanol synthesis pathway could be the possible reason for the ultra-high methanol selectivity.

6. In Fig.4a, the authors declared that “The differential charge density map also evidenced electron density accumulation for the In-H bond and depletion from the hydroxyl group.”. However, it seems that the electron density is almost unchanged for the hydroxyl group.

Author Reply: Thanks for your kind reminder on the presentation of the charge density difference mapping. In a hydroxyl group (OH), the electron density is majorly localized within the O atom where the depletion is expected to be observed. Due to the electron density mapping overlapping with the O atom (please see picture below, **Fig. R2**), we decreased the atomic radius of O in **Fig. 4a** during this revision to clearly show the electron density depletion from the Lewis basic hydroxyl group.

Figure R2. The differential charge density of protonated and hydridic SFLP sites with the decrease of O radius.

7. In the DFT section, the authors declared that “Notably, this energy demand could be fully compensated by the heat release from previous H₂ adsorption and activation steps ($E = -1.93$ and -0.87 eV).”. Such a statement is really unscientific. No research has indicated that the energy released by previous reaction steps can help to overcome the energy barriers of subsequent reaction steps.

Author Reply: The structure transition of In-OH-In to InOH...[O]...In and H₂ adsorption on the In site of InOH...[O]...In are within several atomic distances of SFLP sites. Thus, the released heat during H₂ adsorption could be easily reused before dissipation, as the chemical reaction is known to occur at an fs-to-ns level, which is too short to allow thorough heat dissipation (to a millimeter or an even longer distance) *via* thermal conduction or convection (usually >100 ns; Chen, G. Nanoscale energy transport and conversion: a parallel treatment of electrons, molecules, phonons,

and photons. Oxford University Press (2005)). This is not a DFT problem but a pure thermodynamics problem (the first law of thermodynamics). Supporting papers could be found in J. K. Nørskov's work (*Journal of Catalysis* 182, 479–488 (1999)) and recent advances in ammonia synthesis (*Nature Catalysis* 1, 178-185 (2018); *Chem* 5, 2702-2717 (2019)). Please find the corresponding energy diagrams below.

8. There are even no page numbers for the manuscript and supporting information.

Author Reply: Page numbers have been added in our manuscript and SI.

9. At line 246, the authors stated that “The in-situ DRIFTS was used to track the intermediate species to understand the mechanism of methanol formation on S2 (Figure 4a-b, Figure S21).” Actually, the in-situ DRIFTS spectra were in Fig.3 c, d and Fig.4 a, b were DFT results.

Author Reply: We are deeply sorry for the many mistakes caused by our carelessness. This mistake has been corrected.

10. For the discussion of the in-situ DRIFTS studies, the authors should attribute different peaks to specific chemical groups rather than molecules.

Author Reply: We thank the referee for their valuable suggestions which have helped us improve the quality of the manuscript. We attribute different *in-situ* DRIFTS peaks to specific chemical groups.

The *in-situ* DRIFTS was used to track the intermediate species to understand the mechanism of methanol formation on S2 (Figure 3c-d, Figure S21). The ratio of H₂:CO₂ was 1:1, and the peak at 1070 cm⁻¹ represents methoxy species of C-O stretching vibrations.^{31,41} The observed methoxy

species indicate the formation of methanol. Meanwhile, the formate could either be the source of methanol or CO. To further confirm the methanol pathway and the possibility for tandem methanol synthesis, a 50% H₂ and CO *in-situ* DRIFTS measurement was conducted, **Figure 3d**. Despite the presence of methanol-related peaks at 2800 cm⁻¹ – 2900 cm⁻¹ of C-H stretching^{42,43} and methoxy species of C-H stretching vibrations at 3000 cm⁻¹ – 3100 cm⁻¹,¹⁹ the absence of typical formate species at 3200 cm⁻¹ of **C-H stretching** and 1580 cm⁻¹ of **OCO asymmetric stretching vibrations** imply that the formate species could be the intermediate species for RWGS rather than methanol synthesis. It is worth noting that the peak at 1520 cm⁻¹ could correspond to the **asymmetric HOCO* species stretching vibrations**.

Reviewer #2 (Remarks to the Author):

Many of my original concerns have been addressed, but I still have a few that need to be addressed.

Q1: In their response, the authors state 'Since the flow reactor has a lower reactant to catalyst contact time than the batch system, the reaction performance and kinetic could be different.' and 'Moreover, based on the results from the flow reactor, we can identify the interesting trend of activation energy, the possible mechanism of light and two distinct active sites.' However, the mechanism insight from flow reactor conditions is being presented as general. Based on the significant difference in reaction conditions and activity, it is possible that the reaction mechanism operating under batch and flow conditions are distinct. This should be clarified.

Author Reply: We thank the referee for their valuable suggestions which have helped us improve the quality of the manuscript. We have added "**in flow reactor system**" to the conclusion of the manuscript to emphasize this mechanism.

Q6: The interpretation of the PL data still seems idealized to me and lacks rigor. The sticking point is that the conclusion from 'Thus, we take S1 as the reference, and confirmed that S2 exhibit lower PL intensity, which represents lower recombination efficiency and prolonged lifetime of electron-hole pairs (Figure R7).' is only correct under the assumption that all recombination is **radiative**. I assume that the PL quantum yield of these materials aren't known, but I would expect it to be far lower than 100%. This means that it is safe to assume that there is some level of non-radiative recombination, typical when trapped states are involved in the recombination process. The fact that S2 has lower PL intensity (and a small decrease that appears to be < 5%) at all temperatures does not necessarily mean that the rate of recombination is lower. This data only allows to comment on the rate of **radiative** recombination. This means that a meaningful comment on the lifetime of electron-holes pair cannot be made. For example, the PL intensity of S2 could be lower than S1 because of an increased rate of non-radiative recombination and a shorter electron-hole lifetime.

Author Reply: We thank the referee for their valuable suggestions which have helped us improve the quality of the manuscript. We have corrected our manuscript, and the results are as follows.

To further confirm the possibility of a photochemistry-enabled reaction pathway, temperature-programmed photoluminescence (PL) was conducted. S2 has a lower PL intensity than S1, indicating a lower rate of radiative recombination. Due to the higher concentration of [O] and the

lower rate of radiative recombination, the recombination efficiency of photo-excited electron-hole pairs over S2 could be lower than that of S1³³⁻³⁵ (Figure S15).

Q8: I appreciate the additional UPS experiments to determine the band edges, but I still have concerns about a Tauc plot analysis to extract a band gap. I think it is telling that a difference in band gap of 0.5 eV can be obtained from the Tauc method (2.82 eV was revised to 3.39 eV for S2). To my eye a very wide range of values could be obtained as there is no clear, distinct linear section - the value will be very dependent on the specific range objectively chosen. A key fact to keep in mind is that, as prominently stated in the title, the material is black. A simple band structure with a roughly 3.4 eV band gap is incompatible with a black material. It's clearly absorbing light throughout the visible range, as well indicated in Figure R8/S4. The UPS does seem to indicate a separated VB and CB. The reflectance indicates a wide range of tail/trap states in order to have optical transitions throughout the UV-VIS-NIR spectral ranges.

Author Reply: We thank the referee for their valuable suggestions which have helped us improve the quality of the manuscript.

We agree with the reviewer's comment that the Tauc plot cannot give us an accurate value of the bandgap, and various errors could be made during this process. We believe the UV-Vis-NIR could clearly indicate the strong absorption property of our black indium oxide, and as a photothermal catalyst, the role of a bandgap remains unclear for its reaction mechanism. Thus, to avoid any further confusion, we would like to accept the reviewer's previous suggestion (Q8. Describing a bandgap of 2.82 eV for S2 is unconvincing. The Tauc plot shows a curved trace with no obvious linear region. A bandgap maybe isn't even appropriate for this material) and remove the bandgap part.

Figure R3. The UV-VIS-NIR spectra of S1 and S2.

Q9: There is still the words '(maybe we need a picture or a drawing to illustrate the setup)' in the SI, which is clearly a comment held over from previous edits. There is still no details given about the LED used to excite the samples for the in situ XAS. The wavelength range (or spectral output) of the LED and the irradiation power at the sample should be described.

Author Reply: We sincerely apologize for our careless mistake and thank the referee for their valuable suggestions. The spectrum of the white LED is shown in **Fig. R4**.

Figure R4. The spectrum of a 10W LED lamp.

Reviewer #3 (Remarks to the Author):

The revision is well done in most aspects. However, after the clarifications from the authors and reading through the manuscript, I do not find the methanol selectivity simulation component to be appropriate for inclusion in the final manuscript for the following reasons. I would suggest to remove this component and corresponding derived conclusions.

Author Reply: We thank the referee for their valuable suggestions which have helped us improve the quality of the manuscript.

We have removed the component and corresponding derived conclusions in our manuscript. We only kept the well-known simulation data of thermodynamic equilibrium for methanol synthesis in **Figure S13** to indicate that methanol synthesis is challenging under atmospheric pressure. In addition, the method of ASPEN stimulation has been added to the SI.